# A comprehensive study of metabolite genetics reveals strong pleiotropy and heterogeneity across time and context

Apolline Gallois[1,12], Joel Mefford[2,12], Arthur Ko [3], Amaury Vaysse [1], Hanna Julienne[1], Mika Ala-Korpela[4,5,6,7,8,9], Markku Laakso [10], Noah Zaitlen[2,12]*, Päivi Pajukanta [3,12]* & Hugues Aschard [1,11,12]*

Genetic studies of metabolites have identified thousands of variants, many of which are associated with downstream metabolic and obesogenic disorders. However, these studies have relied on univariate analyses, reducing power and limiting context-specific under-standing. Here we aim to provide an integrated perspective of the genetic basis of meta-bolites by leveraging the Finnish Metabolic Syndrome In Men (METSIM) cohort, a unique genetic resource which contains metabolic measurements, mostly lipids, across distinct time points as well as information on statin usage. We increase effective sample size by an average of two-fold by applying the Covariates for Multi-phenotype Studies (CMS) approach, identifying 588 significant SNP-metabolite associations, including 228 new associations. Our analysis pinpoints a small number of master metabolic regulator genes, balancing the relative proportion of dozens of metabolite levels. We further identify associations to changes in metabolic levels across time as well as genetic interactions with statin at both the master metabolic regulator and genome-wide level.

[1] Department of Computational Biology - USR 3756 CNRS, Institut Pasteur, Paris, France. [2] Department of Medicine, University of California, San Francisco, CA, USA. [3] Department of Human Genetics, University of California, Los Angeles, CA, USA. [4] Systems Epidemiology, Baker Heart and Diabetes Institute, Melbourne, VIC, Australia. [5] Computational Medicine, Faculty of Medicine, University of Oulu and Biocenter Oulu, Oulu, Finland. [6] NMR Metabolomics Laboratory, School of Pharmacy, University of Eastern Finland, Kuopio, Finland. [7] Population Health Science, Bristol Medical School, University of Bristol, Bristol, UK. [8] Medical Research Council Integrative Epidemiology Unit at the University of Bristol, Bristol, UK. [9] Department of Epidemiology and Preventive Medicine, School of Public Health and Preventive Medicine, Faculty of Medicine, Nursing and Health Sciences, The Alfred Hospital, Monash University, Melbourne, VIC, Australia. [10] Department of Medicine, University of Eastern Finland and Kuopio University Hospital, Kuopio, Finland. [11] Department of Epidemiology, Harvard T.H. Chan School of Public Health, Boston, MA, USA. [12]These authors contributed equally: Apolline Gallois, Joel Mefford, Noah Zaitlen, Päivi Pajukanta, Hugues Aschard. *email: Noah.Zaitlen@ucsf.edu; PPajukanta@mednet.ucla.edu; hugues.aschard@pasteur.fr

The human metabolome includes over 100,000 small molecules, ranging from peptides and lipids, to drugs and pollutants[1]. Because metabolites affect or are affected by a diverse set of biological processes, lifestyle and environmental exposures, and disease states[2], they are routinely used bio-markers[3]. Thanks to the recent technological advances, diverse components of the metabolome are being measured in large human cohorts, offering new opportunities to improve our understanding of the molecular mechanisms underlying metabolism and corresponding human traits and diseases[4]. For example, previous work has highlighted the role of metabolites in diseases such as Type 2 Diabetes[5,6], cardiovascular, and heart diseases[7–9], and obesity[3,10].

A number of genome-wide association studies (GWAS) of metabolites have also been performed. These studies identified hundreds of genetic variant–metabolite associations[11], provided estimation of the heritability of multiple metabolites[12], and highlighted the biological and clinical relevance of some of these findings[13]. All of these studies relied on standard univariate analyses and assessed marginal additive effects of genetic variants only. Despite a number of advantages, univariate approaches have limitations and the implementation of new integrative approaches are needed to further reconstruct the complex genetic network of gene–metabolite associations and its dependence on the context of each individual.

Here, we explore the genetics of 158 serum metabolites measured with nuclear magnetic resonance (NMR), including mostly serum lipids, in 6263 unrelated men from the Finnish Metabolic Syndrome In Men (METSIM)[14] cohort. To improve the detection of metabolite-associated variants and infer complex mechanisms underlying the genetics of metabolites, we perform a series of analyses using recently developed multivariate approaches as well as existing methods. First, unlike previous metabolite GWAS[11,12,15–20], we leverage the high correlation structure between metabolites to increase the power of association tests via the Covariates for Multi-phenotype Studies (CMS) method[21]. Second, we produce an integrated view of the genetic–metabolite network, highlighting genes with strong pleiotropic effects while showing how integrated analysis of such genes can be leveraged to identify likely causal variants. Third, we examine variants with effects dependent on statin treatment and age, two established modifiers of metabolite profiles[22] and disease risk, using bivariate heritability and interaction analyses. Overall, our analysis provides a step towards richer understanding of genetic regulation of metabolites as a function of environmental factors.

## Results

**Powerful genome-wide screening.** We first performed GWAS of the 158 serum metabolites. These measurements consisted of 98 lipoproteins components (42 very-low-density lipoprotein (VLDL), 7 IDL, 21 low-density lipoprotein (LDL) and 28 high-density lipoprotein (HDL)), 9 amino acids, 16 fatty acids, and 35 other molecules (Supplementary Data 1 and Supplementary Fig. 1). GWAS was performed using standard linear regression (STD), but also using the CMS approach[21], a powerful method we recently developed for the analysis of multivariate data sets (Online Methods). For both methods, we tested association between each SNP and each metabolite while adjusting for potential confounding factors, including age and medical treatments (statins, beta blockers, diuretics, and fibrate). To approximate the number of independent associations identified, we grouped significant SNPs in independent linkage disequilibrium (LD) blocks[23], denoted further as regions (Online Methods). We obtained 588 region-metabolite associations involving a total of 54 independent regions (Supplementary Data 2, 3). Figure 1a shows that these associations are spread over the 158 metabolites: we found 399 associations with lipoproteins (189 with VLDL, 38 with IDL, 88 with LDL and 84 with HDL), 17 with amino acids, 50 with fatty acids, and 122 with other molecules. Among these associations, 9 were significant with STD only (1.53%), 261 with both STD and CMS (44.39%) and 318 (54.08%) with CMS only (Supplementary Data 4). Overall, CMS led to a 118% increase in identified signals. Among the 588 region-metabolite associations identified, 228 (involving 45 genes) were not identified at the same significance level by previous large-scale metabolite studies[11,12,19,20,24–27] (Supplementary Data 1 and Supplementary Table 6). Note that most of these new associations (78%) involved regions previously identified with total lipids (total cholesterol, triglyceride, LDL, and HDL)[28], but were not further refined into specific particles. As illustrated in Fig. 1b, new associations exist for 107 of the 158 metabolites. Among these 228 associations, 1 was significant with STD only (0.4%), 63 with both STD and CMS (27.6%) and 163 (71.5%) with CMS only. For each new association, we further mapped the top SNPs per region to their nearest gene. Table 1 presents the aggregated results.

Overall, the CMS approach showed good performance in these data. First, when comparing single SNPs effect estimates between

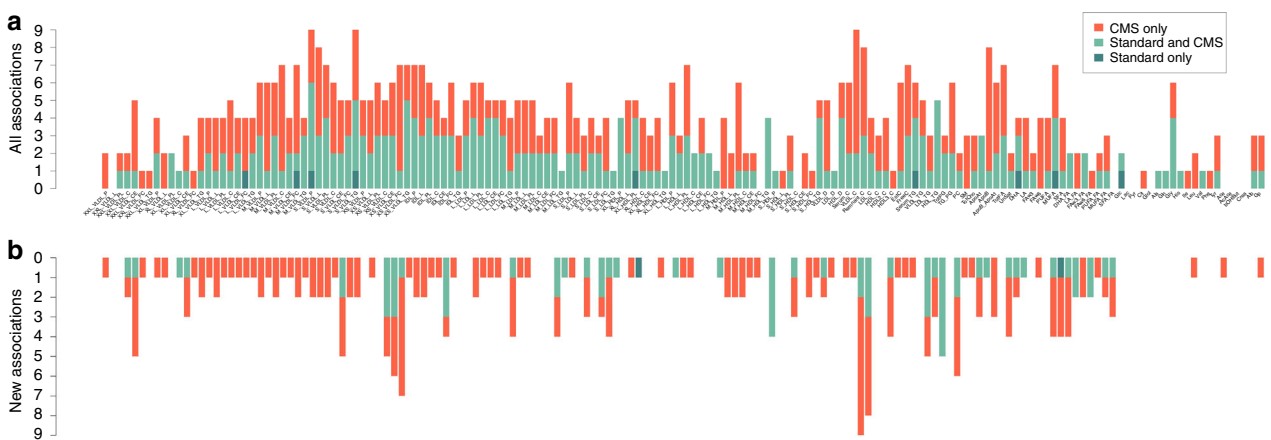

**Fig. 1** Region-metabolite associations. Distribution of the 588 significant associations ($P < 1.28 \times 10^{-9}$) identified in the 158 metabolites GWAS in the METSIM cohort. **a** Regions in dark green were significant for standard linear regression adjusted by confounding factors. Regions in red were significant for linear regression adjusted with confounding factors and covariates selected by CMS. Regions in light green were significant for both models. **b** Same plot including only the 228 new associations, not identified in previous metabolites GWAS

**Table 1 New gene–metabolite associations**

| Chr | Gene[a] | Position | SNP[b] | A1 | A2 | Associated metabolites | Opposite association |
|---|---|---|---|---|---|---|---|
| 1 | PCSK9 | 55,505,647 | rs11591147 | G | T | IDL_CE, L_LDL_TG, M_LDL_FC, Remnant_C, S_LDL_FC, S_VLDL_CE, VLDL_C, XL_HDL_FC, XS_VLDL_C/CE/FC, XXL_VLDL_CE | M_HDL_C/CE/P/PL, S_HDL_PL |
| 1 | DOCK7 | 63,056,112 | rs1748197 | G | A | HDL_TG, MUFA, M_HDL_TG, PC, PUFA, TotCho, TotFA, XXL_VLDL_CE | |
| 1 | CELSR2 | 109,818,530 | rs646776 | T | C | M_LDL_FC, S_LDL_FC/PL | |
| 1 | PSRC1 | 109,822,166 | rs599839 | A | G | S_LDL_CE | |
| 1 | GALNT2 | 230,294,916 | rs2144300 | C | T | ApoB, L_VLDL_*, M_VLDL_*, S_VLDL_FC/L/P/PL/TG, TG_PG, VLDL_C/D/TG, XL_VLDL_P/TG | M_HDL_PL, S_HDL_PL |
| 2 | APOB | 21,225,281 | rs1042034 | T | C | TotFA | |
| 2 | GCKR | 27,730,940 | rs1260326 | T | C | Remnant_C, TG_PG, VLDL_C/TG, XL_VLDL_CE/FC | L_HDL_PL |
| 3 | PROK2 | 71,880,578 | rs7622817 | G | A | Serum_C | |
| 4 | CHIC2 | 54,714,868 | rs17083590 | G | A | XS_VLDL_CE | |
| 4 | UTP3 | 71,552,398 | rs16845383 | A | G | Alb | |
| 5 | MARCH3 | 126,267,351 | rs12655258 | C | T | HDL2_C | |
| 5 | MIR4634 | 174,223,234 | rs12660057 | G | A | M_HDL_L | |
| 6 | MICB | 31,236,410 | rs34131062 | T | C | S_VLDL_TG, VLDL_TG, XS_VLDL_TG | |
| 6 | MIR3925 | 36,613,812 | rs6457931 | G | T | XL_HDL_L | |
| 8 | LPL | 19,832,646 | rs17482753 | G | T | ApoB, HDL_TG, MUFA, SFA, TG_PG, TotFA, VLDL_C/TG | |
| 8 | TRIB1 | 126,485,531 | rs7846466 | T | C | L_VLDL_L, MUFA, Remnant_C, VLDL_C, XL_VLDL_C/CE/L, XXL_VLDL_C/CE/FC | |
| 10 | PCDH15 | 56,015,656 | rs11004183 | G | A | IDL_C/FC/L/P | |
| 10 | PKD2L1 | 102,075,479 | rs603424 | G | A | MUFA_FA | |
| 11 | CELF1 | 47,539,697 | rs4752845 | T | C | ApoA1 | XXL_VLDL_P |
| 11 | PTPMT1 | 47,583,121 | rs12798346 | C | T | HDL_D, L_HDL_P, XL_HDL_PL | |
| 11 | MTCH2 | 47,663,049 | rs10838738 | G | A | TG_PG | |
| 11 | MYRF | 61,551,356 | rs174535 | C | T | PUFA_FA | S_HDL_TG |
| 11 | TMEM258 | 61,557,803 | rs102275 | C | T | MUFA, MUFA_FA | HDL2_C |
| 11 | FADS1 | 61,569,830 | rs174546 | C | T | EstC, FAw3_FA, UnSat, XS_VLDL_L | M_VLDL_FC |
| 11 | FADS2 | 61,597,972 | rs1535 | G | A | DHA_FA, SM, XS_VLDL_FC | LA_FA, M_VLDL_P, XL_VLDL_TG |
| 11 | FADS3 | 61,639,573 | rs174448 | G | A | M_VLDL_PL | |
| 11 | CPT1A | 68,562,328 | rs17610395 | C | T | DHA, DHA_FA, FAw3, FAw3_FA | |
| 11 | APOA5 | 116,660,686 | rs2266788 | G | A | HDL_TG, Ile, M_HDL_TG, PUFA, Remnant_C, SFA, S_VLDL_CE, TG_PG, VLDL_C/TG, XS_VLDL_FC, XXL_VLDL_C/CE | |
| 12 | HNF1A | 121,420,260 | rs7979473 | G | A | M_LDL_P | |
| 13 | LINC02296 | 87,773,653 | rs17123289 | G | A | FreeC | |
| 15 | LOC283665 | 58,380,442 | rs12910902 | T | C | LDL_TG, L_HDL_L, L_LDL_TG | |
| 15 | LIPC | 58,683,366 | rs1532085 | A | G | HDL2_C, HDL3_C, HDL_TG, IDL_CE, LDL_TG, L_HDL_TG, L_LDL_L/TG, MUFA_FA, M_HDL_L/ TG, M_LDL_L/TG, PUFA, Remnant_C, SFA, S_HDL_TG, S_LDL_TG, S_VLDL_C/CE/FC/L/ P/PL, TotCho, VLDL_C, XS_VLDL_C/CE/FC | FAw6_FA, LA_FA, PUFA_FA |
| 15 | MYO1E | 59,453,384 | rs2306791 | T | C | S_LDL_P/PL | |
| 16 | ITGAM | 31,343,769 | rs4597342 | T | C | TG_PG | |
| 16 | CETP | 56,991,363 | rs183130 | C | T | ApoB, HDL_TG, IDL_L/P/PL, L_LDL_C/CE/L/PL, M_HDL_TG, Remnant_C, S_VLDL_CE, VLDL_C, XL_VLDL_CE, XS_VLDL_C/ CE/FC, XXL_VLDL_CE | HDL2_C |
| 16 | DHX38 | 72,144,174 | rs9302635 | T | C | SFA, TotFA | |
| 16 | PMFBP1 | 72,230,112 | rs9923575 | T | C | UnSat | |
| 16 | C16orf47 | 73,177,225 | rs9673570 | A | G | Tyr | |
| 19 | LDLR | 11,202,306 | rs6511720 | G | T | IDL_CE, LDL_TG, L_LDL_TG, M_LDL_FC, Remnant_C, S_LDL_CE/FC, S_VLDL_CE, XS_VLDL_C/CE/FC | |
| 19 | PRKCSH | 11,560,347 | rs755000 | T | G | FreeC | |
| 19 | APOE | 45,408,836 | rs405509 | G | T | M_HDL_P/PL, PUFA | |
| 19 | APOC1 | 45,415,640 | rs445925 | G | A | IDL_CE, M_LDL_FC, S_HDL_CE, S_LDL_CE/FC/PL, TotCho, XS_VLDL_C/CE | |
| 19 | NECTIN2 | 45,373,565 | rs395908 | G | A | Remnant_C | |
| 19 | TOMM40 | 45,395,266 | rs157580 | A | G | VLDL_C, XS_VLDL_FC | |
| 20 | PLTP | 44,545,048 | rs4810479 | C | T | S_HDL_FC/PL | |

*Chr.* chromosome
[a]Nearest gene from the reported SNP
[b]SNP strongly associated with the majority of phenotypes present in last two columns, most significant SNP for each phenotypes are listed in Supplementary Data 5

CMS and STD for the identified association, we observed a very strong correlation (0.99), confirming the absence of bias owing to the adjustment for covariates[29] selected by CMS (Fig. 2a). Second, we plotted the effect size of the SNP as a function of the metabolite variance explained by CMS (Fig. 2b). As expected, the additional associations identified by our method correspond to variants with smaller effect size, captured thanks to increases in statistical power (Supplementary Fig. 2). Third, to graphically

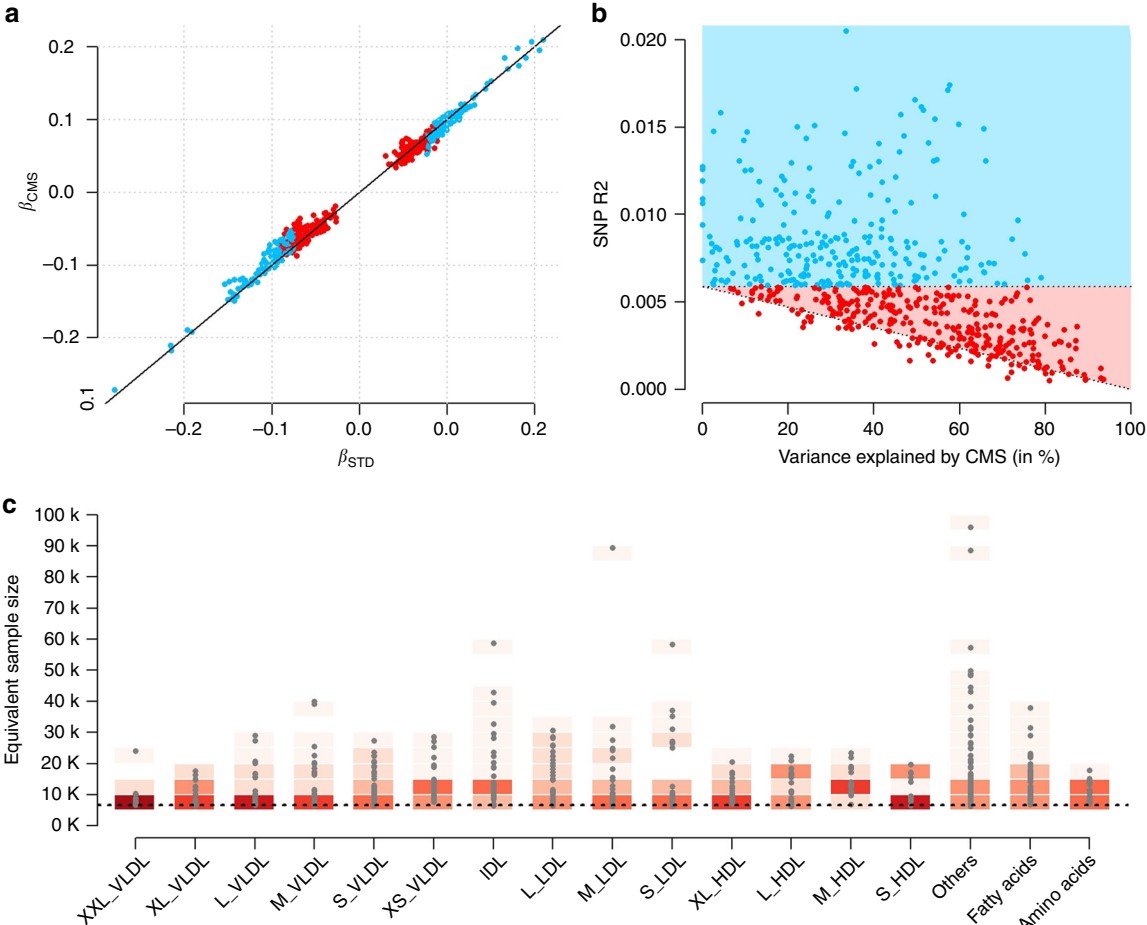

**Fig. 2** Overview of CMS results. Characterization of results from the CMS adjusted analysis among the 588 identified region-metabolite associations. For all panels, we used the most associated SNP per region. **a** The regression coefficient for each SNP estimated using standard linear regression ($\beta_{STD}$) and after adjustment for the covariates selected by CMS ($\beta_{CMS}$). Associations significant at $1.28 \times 10^{-9}$ with the standard test are indicated in blue, those only significant with CMS are indicated in red. **b** The outcome variance explained by the SNPs as a function of the variance explained by covariates selected by CMS for the corresponding associations. The blue and red areas correspond to the detectable SNP effect size for simple regression given the available sample size, and after explaining the residual outcome variance, respectively. **c** The gain in power achieved by CMS across the species analyzed, expressed as equivalent increase in sample size. The dash black line corresponds to the baseline sample size of 6623 individuals. The gradient of reds indicates the density

illustrate the gain in power, we derived the equivalent increase in sample size ($N_{eff}$) achieved by decreasing the overall residual variance (Supplementary Fig. 3), and across the 588 identified associations (Fig. 2c).

The average gain over the ~ 95 million tests performed was modest ($N_{eff} = 8{,}768$). However, for the 588 significant associations, CMS leads to an average $N_{eff}$ of 14,000, which corresponds to a 2.2-fold increase as compared with the baseline sample size of 6263 individuals. The maximum gain in power we observed was equivalent to the analysis of 96,108 individuals (rs174538, in gene *TMEM258*, association with HDL2_C, $P_{CMS} = 6.4 \times 10^{-10}$). Interestingly, that variant, which was only border-line nominally significant in the standard marginal model ($P = 0.058$), was reported to be associated with HDL, LDL, TG, and TC ($P = 7.9 \times 10^{-20}$, $P = 1.1 \times 10^{-34}$, $P = 3.5 \times 10^{-28}$, $P = 2.5 \times 10^{-32}$, respectively) in the Willer et al.[28] study that included 188,577 individuals (Supplementary Data 5).

We next performed in silico replication for all new associations using data from Kettunen et al.[11] ($N = 24{,}925$), the only independent study with available summary statistics for the SNP-metabolites pair we report. Out of the 228 region-metabolite pairs, 88 were available for in silico replication (39%). Among

those, 60 (68.2%) were replicated at a nominal threshold of 5%. Non-replication of the remaining 28 associations is likely explained by limited power in the replication dataset. Indeed, we observed a strong correlation ($\rho = 0.63$) between the effect sizes of top SNP per region derived from METSIM and the strength of signal for the same variant in the Kettunen et al.[11] study (Supplementary Fig. 4).

Eventually, when comparing the top SNPs from every region associated with at least one metabolite ($N = 70$, see next paragraph) with previous GWAS on coronary heart disease (CHD)[16], body mass index (BMI)[17], and type 2 diabetes (T2D)[18], we observed substantial enrichment for nominally significant association. Given a false discovery rate (FDR) at 10%, we observed 30 significant genes for CHD, 5 for BMI, and 4 for T2D (Supplementary Data 6 and Supplementary Note 1), indicating some of these variants are also involved in the genetics of common diseases.

**Master regulators of lipids.** We observed substantial evidence of polygenicity and pleiotropy. Using the aforementioned SNP-gene assignment, 147 metabolites were associated with at least one

gene, and a total of 70 genes associated with at least one meta-bolite. Metabolites were associated with one to nine genes, with an average of four genes. On the other hand, genes showed high level of pleiotropy with an average of 8.4 metabolites associated with each gene. We found that 13 master metabolic regulator genes (*LIPC, APOA5, CETP, PCSK9, LDLR, GCKR, APOC1, LPL, GALNT2, CELSR2, TRIB1, DOCK7,* and *FADS2*) capture over 75% (N = 457) of all associations (Supplementary Fig. 5). As mentioned previously, those are the nearest genes to the top associated variants for each region. For clarity, we use those genes throughout our study, however, this list should be considered with caution as the genetic effects of the associated variants might potentially be attributed to other genes. For example, we performed a bioinformatics analysis using FUMA[30], mapping

variants with genes based on their association with gene expression. For many regions, the variants in questions were associated with a range of other candidate genes besides the listed ones (Supplementary Data 7).

The extensive pleiotropic effects in this regions are illustrated in Fig. 3, which includes all associations plotted in a Cytoscape[31] network. The network highlights several known master regulatory effects of genes. For example, *CETP* encodes a protein that transports cholesterol esters and triglycerides between HDL metabolites and VLDL metabolites. Our network clearly displays the opposite effect of variants in *CETP* on HDL and VLDL. Our results also contribute explaining the complex effect of *PCSK9*. Besides its established association with LDL and VLDL, our analyses confirm opposite associations with HDL metabolites[32].

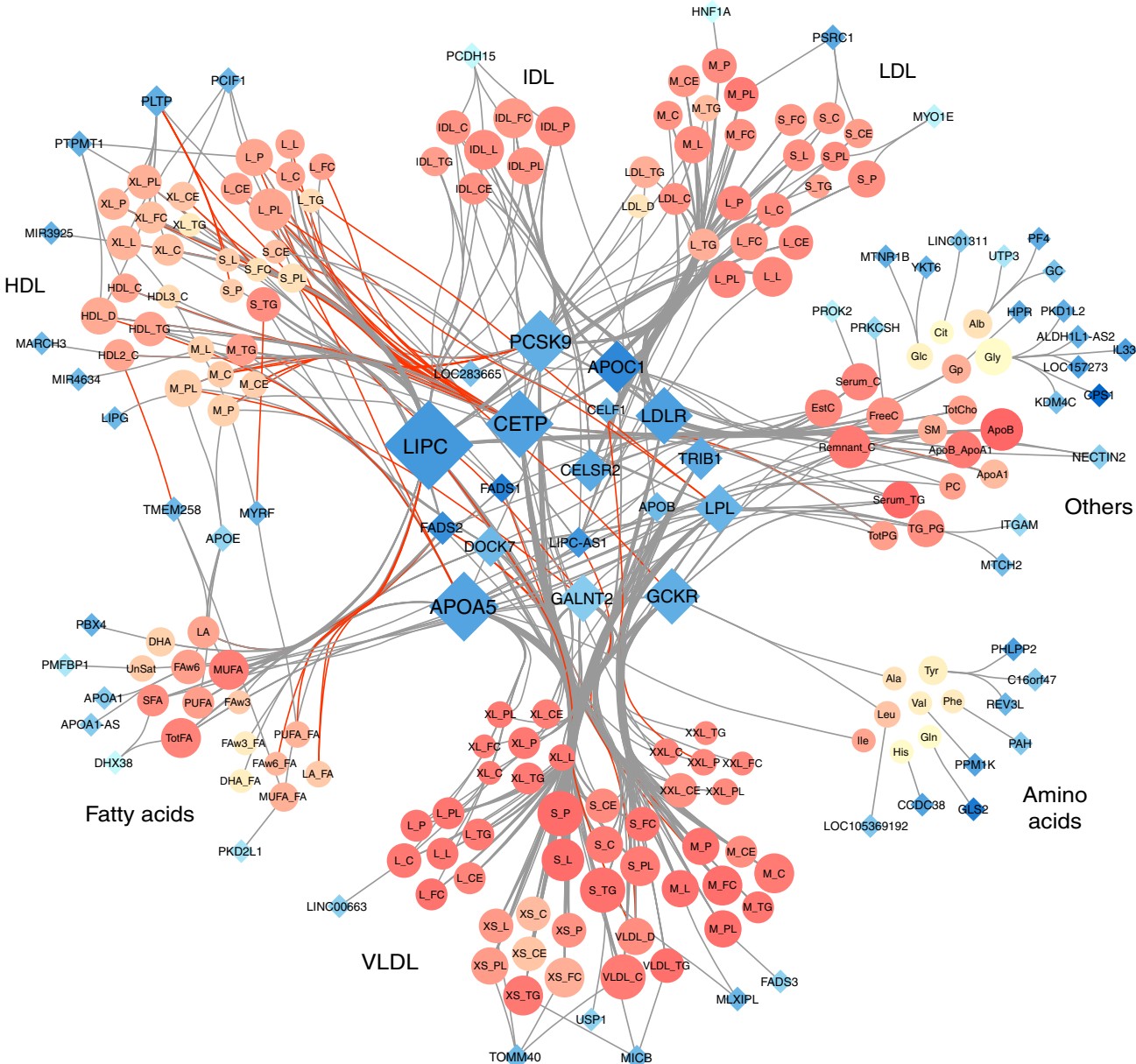

**Fig. 3** Network representation of the 588 region-metabolite associations identified in the 158 metabolites GWAS in METSIM. For each region we used the nearest gene of the most associated variant. Each node represents either a gene (blue diamonds, N = 70) or a metabolite (orange circles, N = 147). Each edge is an association between one gene and one metabolite. Node size is directly proportional to the number of other nodes associated with it. Red edges correspond to opposite effect of a gene on a metabolite, compared with the other metabolites associated with the same gene. Metabolites colors (orange shades) represents correlation strength between a given metabolite and all other metabolites. Genes colors (blue shades) represent strength of correlation between a given gene and associated metabolites, quantified as the average of $r^2$ across all corresponding metabolites

Overall, the gene displaying the strongest pleiotropic effect was *LIPC* with 75 associated metabolites, of which 34 were new associations (11 of them were available for replication, and 8 were replicated at a 5% *p* value threshold).

To better understand the role of these master regulators we performed two additional analyses. First, we appreciate that the observed pleiotropy for these genes is relative, because of the strong correlation across phenotypes. To approximate the number of independent components associated with each gene, we derived the number of principal components (PCs) necessary to explain percentages of the total variance of the corresponding associated metabolites (Supplementary Table 2). Overall, although there is, as expected, a decrease in the total number of independent components, the number of potential meaningful association remains quite high. It required on average 10%, 30%, and 50% of the PCs to explain 90%, 99%, and 99.9% of the total variance, respectively. There was limited variability across genes, and similar numbers were observed when focusing only on lipoproteins. For example, for *LIPC*, it required 19 and 12 PCs to explain 99% of the variance of the 75 metabolites, and the 43 lipoproteins, respectively.

Second, we synthesized the results across the lipoproteins, which contribute to the majority of the observed associations (Supplementary Methods and Fig. 4). Overall, the genes show homogeneous association by lipoprotein class (particles, lipids, phospholipids, cholesterol, cholesterol ester, free cholesterol, and triglyceride), some variability by size (extremely large, very large, medium, small, and very small), and strong heterogeneity by type (VLDL, HDL, LDL, and IDL). We observed three major patterns: (i) *CETP*, *FADS1-2*, *DOCK7*, and *LIPC* are mostly associated with VLDL and HDL, but with differences in the size of the associated lipoprotein: *FADS1-2* and *DOCK7* are enriched for association with very large and medium size, respectively, while *CETP* and *LIPC* displays association with lipoproteins of all sizes; (ii) *TRIB1*, *LPL*, *GCKR*, *GALNT2*, and *APOA5* are mostly associated with VLDL of average size; and (iii) *PCSK9*,

*LDLR*, *CELSR2*, and *APOC1* are associated primarily with large and lipoprotein LDL. Looking at other lipoprotein associated genes, we found that some might fit in the second categories, but a majority appears to have more-targeted effects, being associated with specific types and sizes (Supplementary Note 2 and Supplementary Fig. 6).

**Fine mapping of LIPC leveraging pleiotropic effects.** For pleiotropic genes, we observed heterogeneity in the number of reported top SNP across metabolites (Fig. 4b). For example, the top SNP for *APOC1* was the same across all 33 associated metabolites (rs445925). Conversely, there were nine top SNPs for the 75 metabolites associated with *LIPC*. Part of this heterogeneity might be explained by LD in these regions, but also by the presence of multiple causal variants affecting different metabolites. To investigate this possibility, we applied the FINEMAP[33] algorithm using the example of the latter *LIPC* region after performing additional genotype imputation in that region (Supplementary Methods and Supplementary Data 8). Our analysis suggests there are at least three distinct association sites with consistently high probabilities of causal effects from seven SNPs and heterogeneous metabolite association patterns, confirming the likely presence of metabolite-specific variants within this gene (Fig. 5 and Supplementary Table 3 and Supplementary Data 9).

We cross-referenced top variants of these three sites with GWAS of common human diseases[34], and functional annotations from *Haploreg*[35]. The first site (A) is composed only of SNP rs10468017, which was previously found associated with age-related macular degeneration (AMD)[36–38] and with *LIPC* expression in human liver tissue[39]. The second site (B) includes four SNPs in complete LD that were previously associated with hypertension[40] and AMD[41,42]. Among the four SNPs, rs2070895 is the strongest candidate in our data with a potential association path through a regulation by *USF1*, a gene with a record of association with lipids[43–46]. Finally, the last site (C) included two

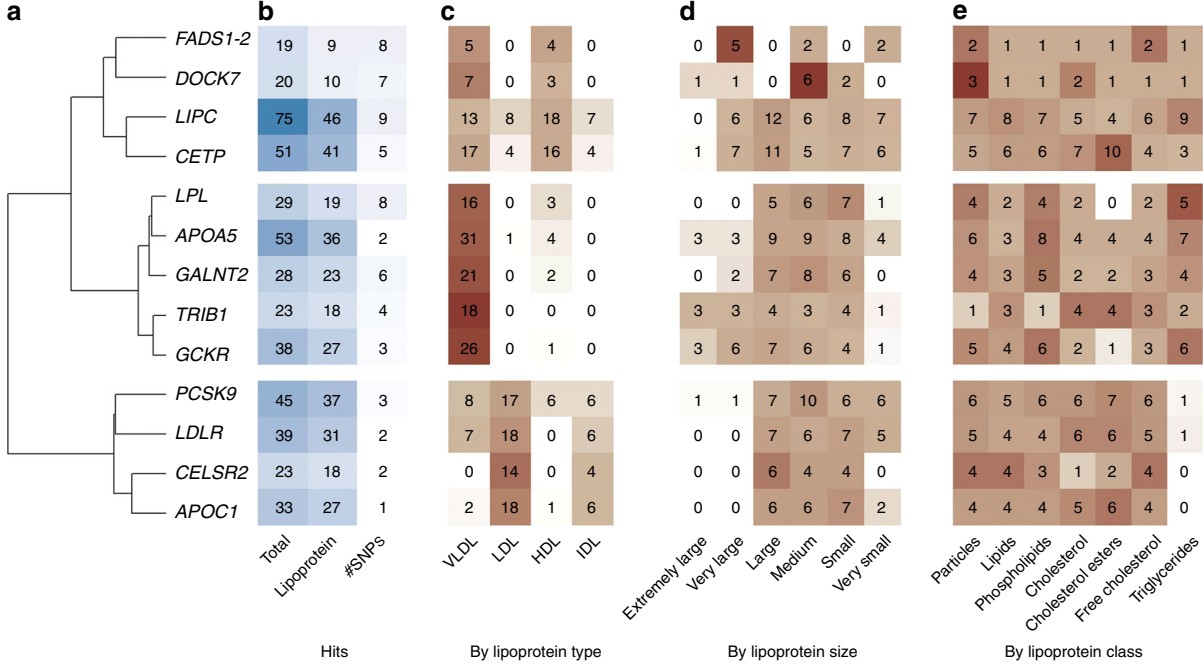

**Fig. 4** Specificity of master regulators on lipoproteins. We performed a hierarchical clustering of the association between the 13 master regulators and the lipoprotein type (**a**). Further panels show the total number of associations, the number of associations with lipoprotein, and the total number of top associated SNP (**b**); the count of association hits by lipoprotein type (**c**), their size (**d**), and class (**e**). The background colors represent the relative proportion of association within each gene-item stratum, highlighting heterogeneity in the distribution of signal

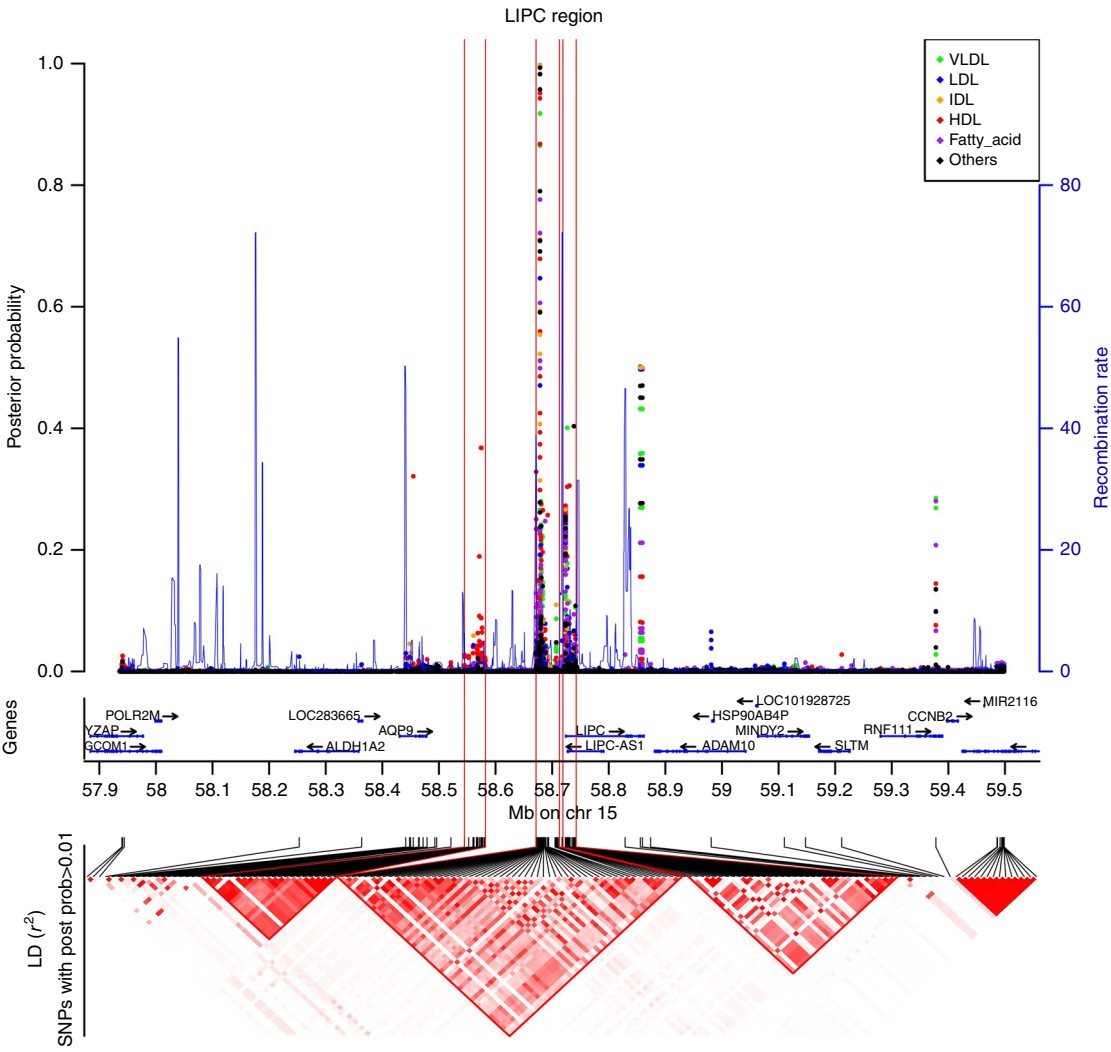

**Fig. 5** Fine mapping of the *LIPC* region. The top panel indicates the posterior probability assessing the evidence that the SNP is causal for each of the 75 phenotypes and the local recombination rate. The middle panel contains genes from the UCSC hg19 annotation. The lower panel is a $r^2$-based LD heatmap computed using PLINK1.9 on the METSIM data. The gradient of red is proportional to the $r^2$. For clarity, we represented the LD only for SNPs with a posterior probability > 0.01 for at least 1 phenotype

SNPs, among which rs113298164 clearly harbored the highest number of relevant bio-features. It is a rare missense mutation, which has been reported to be involved in hepatic lipase deficiency[47]. Additional details on the functional annotation analysis are provided in the Supplementary Note 3.

Deciphering the posterior probability across all SNP-metabolite pair would be challenging because of the dimensionality of the fine-mapping results. However, some global patterns were observed (Supplementary Data 9). Overall, large HDL (L_HDL), and triglyceride in lipoprotein (L_HDL_TG, IDL_TG, L_LDL_TG, XL_HDL_TG, LDL_TG, S_LDL_TG, M_LDL_TG, HDL_TG, XS_VLDL_TG) appear to be influenced by all three likely causal sites. Conversely, intermediate-density lipoproteins (IDLs) and several fatty acids (SFA, PUFA, FAw6, TotFA) are likely mostly influenced by sites A and B. Very small VLDL (XS_VLDL) also display heterogeneous posterior probabilities, highlighting mostly variants from site B as likely causal. Finally, although the three sites showed the highest posterior probability for most of the metabolites, other variants in the region might be involved. For example, an additional variant (rs7177289) displays the strongest posterior probability for the ratio of fatty acids (FAw6_FA, MUFA_FA, LA_FA, and PUFA_FA).

**Dependence of genetic effect on statin use**. An important component of the METSIM cohort is the collection of statin use among participants. To examine changes in genetic regulation of metabolites when taking statins, we performed an interaction test between SNPs and statin for each of the 588 region-metabolite associations, including the 457 associations with the 13 regulator genes. Although no interaction test passed a Bonferroni correction for multiple testing (i.e., $p < 8.5 \times 10^{-5}$, Supplementary Data 10), 83 out of the 588 region-metabolite association showed nominally significant interactions (i.e., $p$ value < 0.05). Based on the $q$ value distribution[48], there were 35 significant interactions at a 10% FDR, showing that at least some of the identified individual SNP-metabolite effects depends on statin use status. Most of these interactions were observed for *APOC1* and *TRIM1* genes, whereas other genes (*FADS1, FADS2, MARCH3, MIR3925, MIR4634,* and *ITGAM*) show interaction with a single metabolite. We also checked statin interaction in follow-up data (see Online methods), and found limited interaction values, except for *APOC1* region, in which 90% of interaction signals found in baseline data were replicated at the 5% significance threshold.

Several of the 13 identified master metabolic regulator genes showed enrichment for negative interaction effects (Fig. 6a).

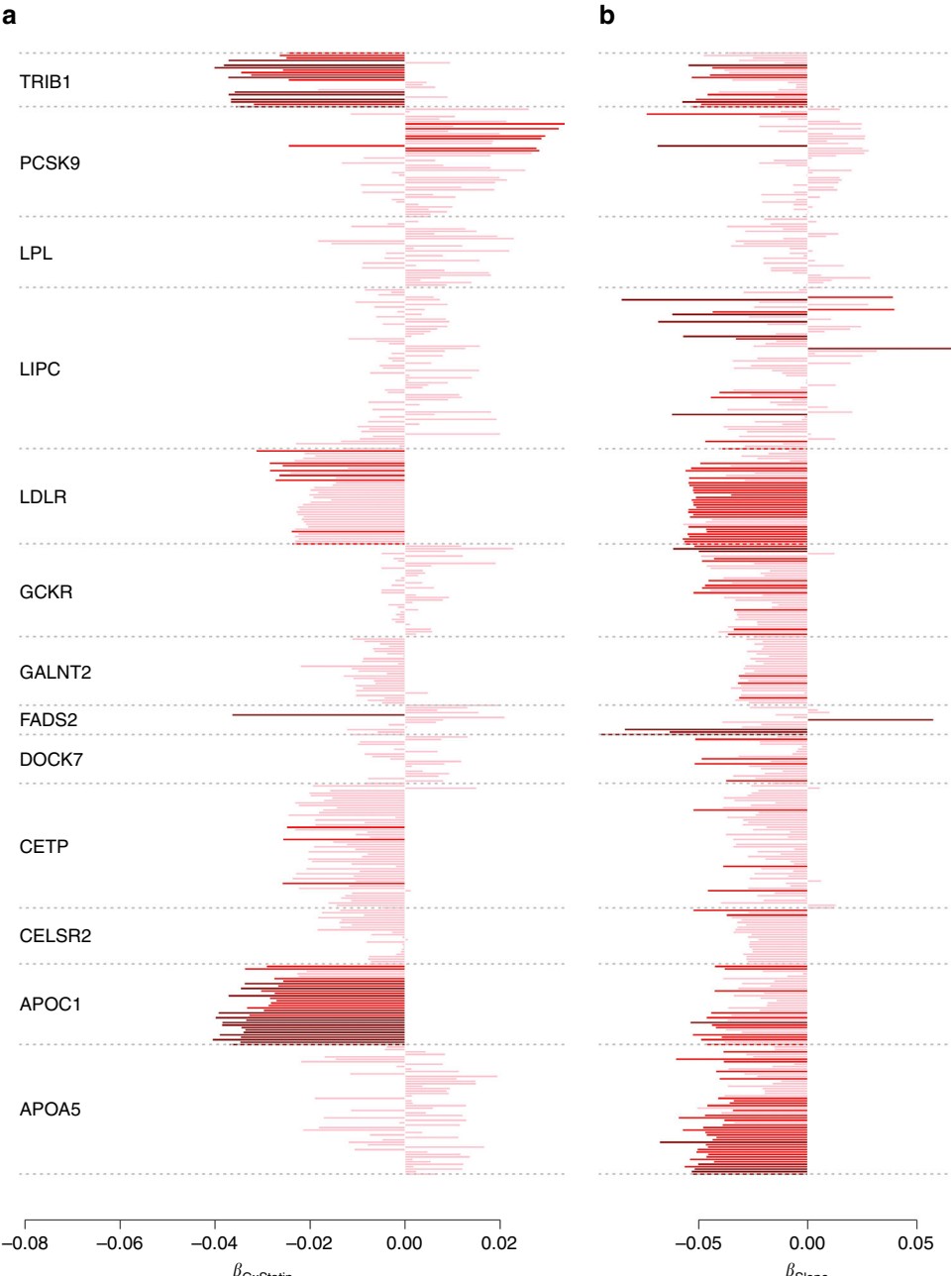

**Fig. 6** Change in genetic effect as a function of aging and exposure to statins. We derived for the top variant of each of the 13 core regulator genes we identified, **a** the interaction effect with statin, and **b** the effect on $\Delta_{bf}$, the difference in the metabolite measurement between the two time points. In all association tests, the allele associated with an increase level of metabolite in the marginal test at baseline measurement was defined as the coded allele. Non-significant test are in pink, test nominally significant are in red, and test significant at $5 \times 10^{-3}$ are in dark red. In agreement with the heritability analysis most of the coefficients for $\Delta_{bf}$ are negative, indicating an overall decrease of genetic effect. Two genes, *APOC1* and *TRIB1*, show strong enrichment for interaction with statin

To account for the correlation between the associated phenotypes, we performed trend tests where individual interaction statistics where merged through a linear combination of single metabolite interaction statistics (Supplementary Table 4 and Online Methods). The analysis confirms the strong significance of the enrichment for negative interaction with statin for *APOC1* ($P = 5.3 \times 10^{-7}$, mostly associated with LDL and IDL particles), *TRIB1* ($P = 1.4 \times 10^{-4}$, associated with VLDL particles), and *LDLR* ($P = 2.4 \times 10^{-4}$, associated with VLDL, LDL, and IDL), and nominal significance for *CETP* ($P = 0.026$, mostly associated with VLDL and HDL). Interestingly, previous work showed association

between *APOC1* and statin-mediated lipid response[49], and statin was also shown to be associated with an upregulation of the expression of *LDLR*[50]. Overall, our analysis suggests that the effect of statin on metabolites might be modified by only a few core genes, and that these interactions do not only impact LDL, but also a range of other lipids.

**Change in genetic effect with age.** Another unique aspect of the METSIM cohort is a second measurement of the same metabolites, using the same technology, ~ 5 years after the baseline

(Online methods) for 3351 unrelated individuals. We used these data to screen for genetic variants associated with an intra-individual change in metabolites level across time. We applied the same strategy as for our primary analysis but using the difference between follow-up and baseline data divided by age difference as outcome ($\Delta_{fb} = (f - b)/(age_f - age_b)$), whereas adjusting for the same confounding factors as baseline and covariates selected by CMS in baseline measurements. There were 30 SNP-metabolites pairs reaching the standard $5 \times 10^{-8}$ p value threshold with either STD or CMS (Supplementary Data 11), corresponding to eight region-metabolite associations (Supplementary Data 12). To the best of our knowledge, these are the first reported SNPs associated with changes in metabolic activity during aging. These associations involved seven metabolites: S-HDL-TG, VLDL-C, DHA, DHA/FA, LA/FA, Faw3/FA, FAw6/FA, and six genes: *PDZRN4*, *LGMN*, *FADS1*, *FADS2*, *TNIK*, *LIPC*. Four of these associations were genome-wide significant in the marginal association at baseline ($P < 5 \times 10^{-8}$). The four new signals were observed for S_HDL_TG, VLDL_C, LA_FA and Faw6_FA. We also performed age interaction test on the linear regression between $\Delta_{bf}$ and significant SNPs (Online methods). However, none of the age interaction p values was significant.

As for the SNP-by-statin interaction analysis, we observed strong concordant effects for the 13 master metabolic regulator genes (Fig. 6b), and performed trend tests to assess the significance of these results. Our analysis showed a strong enrichment for negative genetic effect on all associated metabolites, with 10 out of 13 genes showing nominal significance for an overall decrease of genetic effect with the difference in metabolite between the two time points. The strongest decrease was observed for *APOA5* ($P = 4.6 \times 10^{-6}$) and *LDLR* ($P = 3.0 \times 10^{-5}$). The three genes unaffected were *LIPC*, *LPL*, *PCSK9*, suggesting that the effect of these genes remain persistent with aging, although the relative importance of *LIPC* across some associated metabolites might be affected. For example, *LIPC* showed strong positive association with $\Delta_{bf}$ of the Ratio of omega-6 fatty acids to total fatty acids ($P = 1.6 \times 10^{-7}$) and a decrease in Triglycerides in medium HDL ($P = 2.4 \times 10^{-6}$).

Finally, to examine global changes of genetic regulation of metabolites across time we also estimated heritability for each phenotype at each time point as well as the genetic and environmental correlations of the same phenotype between time points using bivariate linear mixed models[51,52]. Figure 7 and Supplementary Data 13 give heritability values for each metabolite, in both baseline and follow-up data. To avoid any bias in heritability estimation, we computed it on samples present in both baseline and follow-up studies and excluded those who were present in baseline study only. The average heritability decreased from 24.9% at baseline to 18.8% at follow-up, with only 30.8% (p value < 2e-9) having higher heritability at follow-up. The sample size was not large enough to estimate genetic correlation with low standard error, but the average estimate of 0.92, and the strong correlation of fixed effect sizes between time points (Supplementary Table 5), suggests that increasing environmental variance as opposed to decreased genetic variance underlie the reduction in heritability. If true, this result might also explain the absence of SNP-by-age interaction signal in our previous analysis.

## Discussion

Metabolites have been implicated as important factors in many human diseases[3,5–10] and identifying the genetic variants controlling circulating metabolites and their relationship to clinical and environmental characteristics is one of the many challenges facing the human genetics community. Here, we address these questions for the analysis of 158 metabolites measured in more

than 6263 individuals. Our study identified a large number of new region-metabolite associations and highlighted a small number of master metabolic regulator genes that likely play a role in balancing the relative proportion of circulating serum lipids. Indeed, 75% of the 588 identified gene-metabolites association only involved 13 genes, with the top one, *LIPC*, being associated with 75 metabolites. We further showed that genetic effects on metabolites in general, and of these core genes in particular, is modified by statin and aging. More precisely, SNP-by-statin interaction highlighted three genes, *APOC1*, *TRIB1*, and *LDLR*, as modifiers of the statin effect on lipids. Two of them, *APOC1* and *LDLR*, have already been discussed in previous work as candidates for varying the magnitude of statin-mediated reduction in total and LDL-cholesterol. As for aging, all analyses we conducted pointed toward an increase of the environmental variance, leading to a decreased role of genetics among older individuals.

Our study introduces several novelties. We performed a large-scale application of the CMS method we recently developed. For each metabolite-SNP association test, the approach selects additional metabolites that can be used as covariates in a standard linear model, in order to reduce the residual variance. In these data, CMS resulted in an average power gain equivalent to a 1.4-fold increase in sample size. However, among the identified associations, the average gain corresponded to a 2.2-fold increase (i.e., to an effective sample size of ~ 14,000 individuals). In the most extreme case, the gain in power was equivalent to the analysis of > 96,000 individuals, thus demonstrating the strong potential for this approach in future studies. We also performed a large-scale genome-wide study exploring genetic effect on change in circulating metabolites between two time points, providing both individual SNPs GWAS and co-heritability results derived using a bivariate linear mixed model applied to individual-level data.

To better understand the role of the master regulators in the etiology of lipoprotein components, which contributed the vast majority of the reported associations, we performed a series of analyses that highlighted a limited number of association patterns. Overall, three genes *LIPC*, *LDLR*, and *PCSK9* showed global effects, and were associated with all types and sizes of metabolites. On the other hand, some equally pleiotropic genes (in terms of total association reported), such as *APOA5* and *GCKR*, appear to affect specific types of lipids, and VLDL in particular. Many of the master regulator variants also display different top associated variants depending on the metabolites analyzed. Although some of this variability might be owing to LD in these regions, our fine mapping of the *LIPC* region showed that the pleiotropic effect of at least some of these genes is likely owing to multiple genetic variants with heterogeneous effects on the associated metabolites.

More systematic fine mapping of all identified associations is out of the scope of this work as it would require high-density SNPs data including rare variants. Future fine-mapping studies may further improve resolution by leveraging functional annotations[53,54]. Moreover, the fine-mapping analysis we performed for *LIPC* demonstrated how multi-trait associations might help identify likely causal variants. However, we used a naive approach that simply aggregates univariate fine-mapping results. More advanced methodologies integrating all information into a single framework could provide more-accurate posterior probabilities on likely causal variants.

Finally, the observation of age and statin interactions further highlights the utility of obtaining extensive clinical phenotype data as well as collecting multiple time points, which are much better powered to identify age effects than cohort studies. The statin interactions suggest that genetic variation may influence the effectiveness and impact of the drug at a given dose, and may underlie our recent observation[55] that statin effects are

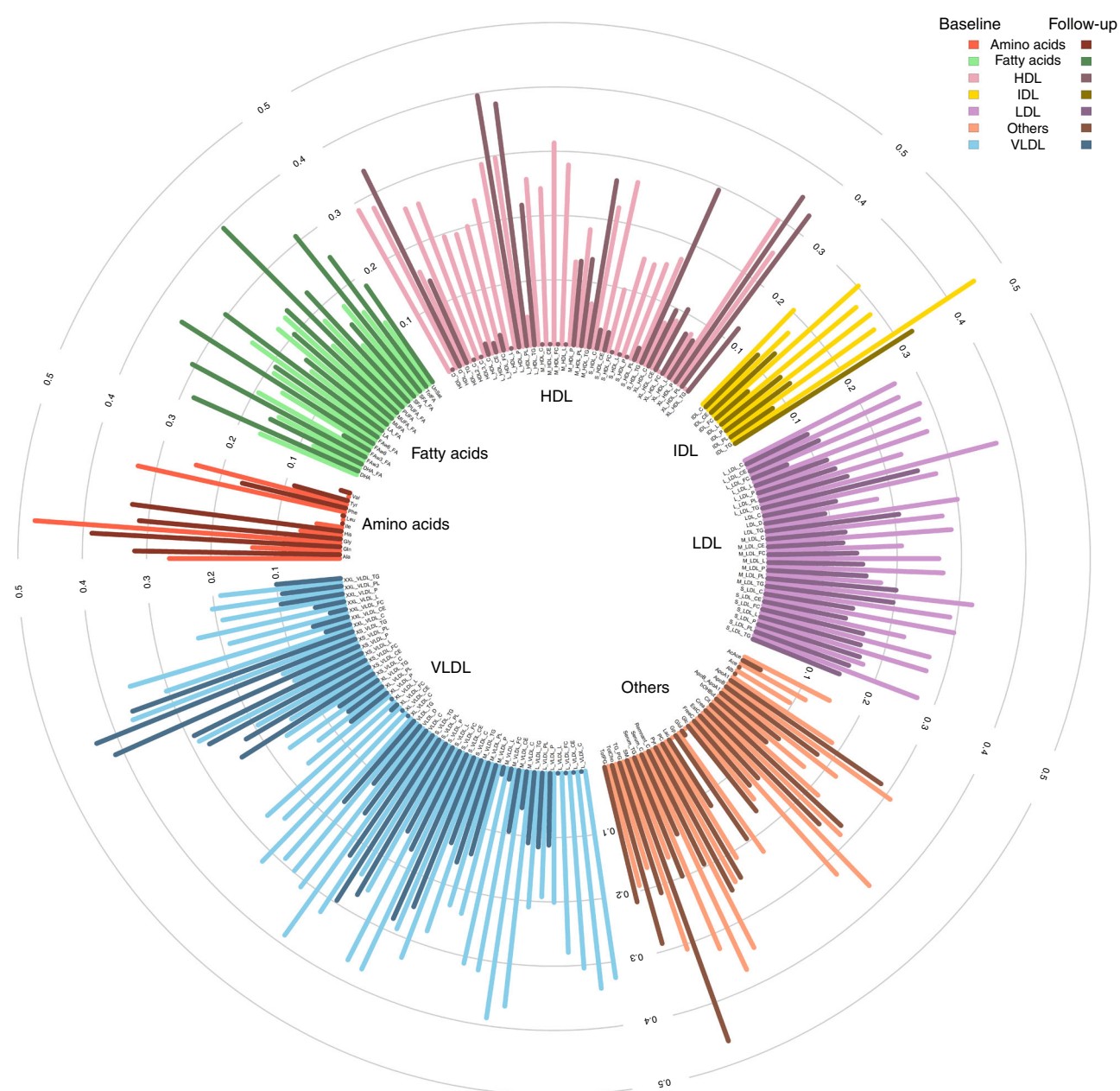

**Fig. 7** Heritability of metabolites in baseline and follow-up data. Heritability of studied metabolites, computed on individuals present in both baseline and follow-up data. We used bivariate restricted maximum likelihood (REML) and included 10 genetic PCs, age, and age² as fixed effects. Light colors stand for heritability in baseline data and dark colors stand for follow-up data

non-uniform on secondary phenotypes such as fasting glucose across individuals. Although the current study is not sufficiently powered to examine these questions directly, it does identify relevant genes to examine for pharmacogenomic studies of statin in properly designed cohorts. The age interactions are also a unique aspect of this work and raise the possibility that genetics can impact trajectories of metabolism over an individual's life span. Although speculative, the most intriguing possibility is that genetic variants could mitigate metabolic disease risk by slowing the natural alteration of metabolic profiles across time.

There are several other shortcomings of this work. First, the CMS approach is currently limited to the analysis of unrelated individuals, and has a much higher computational cost than standard linear regression. As a result, we had to remove related individuals from our analysis and limit the screening to the 600 K

genotyped variants. We are currently developing an improved implementation of CMS addressing these limitations. This will allow for further increases in power thanks to the addition of related individuals, and the analysis of imputed genotypes that might help refine signal at associated regions. Second, TWAS analyses would be of high interest to further explain the link between genetic variants and circulating metabolites. However, TWAS estimates[56] were not available for many of the core metabolic genes, but they could become feasible as larger RNA-seq data sets across more tissues are produced. Third, our metabolite panel included mostly serum lipids, and the presence of a limited number of master regulators only apply to those phenotypes. Whether other sets of related metabolites have a similar genetic architecture, or even share the same regulators, would have to be determined in cohort with a broader range of

metabolites. Fourth, we mapped associated variants with the closest gene. However, extended fine-mapping analysis, as mentioned above, as well as gene expression analysis across multiple tissues might demonstrate a mode of action through other genes.

In conclusion, our study shows that the genetics of lipid metabolites is strongly interconnected, harboring core regulator genes with strong pleiotropic effects, and that other metabolite-associated factors might interfere with some of these core genes, decreasing or increasing their overall effect on metabolites. Further characterizing such global effects would be of particularly high interest in the assessment of drug treatments targeting metabolites. Finally, with the increasing amount of genomic data available, our study as well as previous work[17,57], demonstrated the importance of developing and implementing novel approaches and analytical strategies that allow for a more extensive use of the data and to move toward a more integrated perspective on multivariate molecular phenotypes.

## Methods

**METSIM cohort**. The METSIM cohort[14] is composed of 10,197 Finnish men from 45 to 73 years old and aimed at investigating non-genetic and genetic factors associated with Type 2 Diabetes and cardiovascular diseases. Participants were recruited and examined between 2005 and 2010 in Kuopio town in Eastern Finland. The study was approved by the ethics committee of the University of Kuopio and Kuopio University Hospital, in accordance with the Helsinki Declaration. All study participants gave written informed consent. For each sample, 228 serum metabolites (lipids, lipoproteins, amino acids, fatty acids, and other low molecular weight metabolites) measurements were made with NMR at baseline. A follow-up study was conducted ~ 5 years after the baseline study. In all, 6496 participants (64%) were reexamined with the same protocol and metabolites were measured a second time using the same technology. In our study, we considered 158 variables, including 150 raw measurements and eight ratios. Other available variables, which were mostly percentages, were not included in the study. Besides metabolic measurements, several variables were also available including drug treatment and large group of other phenotypes. All samples were genotyped for 665,478 SNPs using the *Illumina OmniExpress* chip. Genotype data went through standard quality control, filtering individuals with missing rate below 5%, and SNPs with missing rate below 5% or with $P < 10^{-5}$ in Hardy–Weinberg test.

**Metabolites profiling**. We used a high-throughput serum NMR platform for metabolic profiling. Details of this platform have been published previously[58,59] and it has been widely applied in genetic and epidemiological studies[60,61]. This refined targeted metabolomics panel of > 100 serum metabolic phenotypes, includes lipoprotein subclass and lipoprotein lipids, fatty acids, and amino acids assessed by NMR from serum samples. Overall, 14 lipoprotein classes varying in sizes were analyzed including six classes of VLDLs, one class of IDLs, three classes LDLs, and four classes of HDLs. Within each lipoprotein particle, the concentrations of the following lipids were measured: total lipids, phospholipids, total cholesterol, cholesterol ester, free cholesterol, and triglycerides.

**Data pre-processing**. In order to remove outliers without reducing sample size, we first applied inverse normal rank-transformation on every analyzed metabolite. This was done using the *rntransform* function in R package GenABEL[62]. Because of potential confounding effect of statins use on metabolites, we excluded all statins users (1722 individuals) when analyzing LDL, IDL, Apolipoprotein B and cholesterol. We also excluded fibrates users (25 individuals) when analyzing VLDL, IDL, triglycerides, and chylomicron for similar reason. Finally, we removed all individuals with a genetic relationship coefficient larger than 0.05 and used only unrelated individuals. After filtering, there remained 6263 samples available for analysis. For SNP data, we filtered variants with a minor allele frequency lower than 1%. In all, 609,262 SNPs remained after filtering.

**Genome-wide association screening**. We used two different models in the analysis. First, we performed an STD between each metabolite ($Y$) and each SNP ($G$), adjusted for established confounding factors ($C$): age and medical treatments (statins, diuretics, fibrate, and beta blockers):

$$Y \sim \beta_G G + \boldsymbol{\beta_C} \mathbf{C} \tag{1}$$

Then, we used the CMS algorithm to select additional covariates for each SNP-metabolite pair tested. Consider a metabolite $Y_k$, which we refer further as the primary outcome. The CMS approach select potential covariates from the set of available metabolites $Y_{l \neq k}$. In brief, the algorithm is divided in four steps. The first step is the computation of marginal effects through standard linear regressions between variables taken two by two: (i) $Y_k \sim G$ where $G$ is the genetic variant tested, (ii) $Y_{l \neq k} \sim G$ where $l$ includes a subset of candidate covariates (see next

paragraph), and (iii) $Y_k \sim Y_{l \neq k}$. The second step consists in filtering covariates based on a multivariate test between $G$ and all $Y_{l \neq k}$. In practice, it uses a Multivariate analysis of variance (MANOVA), which is applied iteratively, removing one by one covariates potentially associated to the genetic variant tested, until $G$ does not display association with $Y_{l \neq k}$ in the MANOVA. The third step is the filtering of covariates based on $Y_{l \neq k} \sim G$ association conditional on $Y_k \sim G$ association (see Supplementary Methods). It is a stepwise procedure that removes progressively covariates that are potentially associated with $G$. The last step consists in a linear regression between predictor and outcome, adjusted for the selected covariates ($\mathbf{Y_L}$):

$$Y_k \sim \beta_G G + \boldsymbol{\beta_C} \mathbf{C} + \boldsymbol{\beta_L} \mathbf{Y_L} \tag{2}$$

To address some of the limitations of CMS, we also applied for each outcome $Y_k$ a pre-filtering of candidate covariates $Y_{l \neq k}$ before applying CMS. First, to avoid bias owing to very high correlation between covariates and the outcome, we excluded all $Y_{l \neq k}$ explaining > 70% of the outcome variance. Second, to reduce the risk of false positive owing to the inclusion of covariates that are hierarchical parent of the outcome under study, we excluded from the set of initial covariates all secondary outcome that were in the same biological group (LDL, HDL, …) as the primary outcome. Third, to reduce the computational burden, we reduced the number of candidate metabolites used as input of CMS to 30 through on AIC (Akaike information criteria, Supplementary Methods, Supplementary Figures 7–9). As showed in Supplementary Fig. 7, it allows reducing substantially the computation time, while focusing on candidate covariates that altogether still explain a substantial proportion of the primary outcome variance.

All reported $p$ values, whether for marginal genetic effect, or interaction effect have been derived using a Wald test, –i.e., $t = \hat{\beta}^2 / \hat{\sigma}_{\hat{\beta}}^2$, where $\hat{\beta}$ is the estimated regression coefficient, $\hat{\sigma}_{\hat{\beta}}^2$ is the estimated variance of $\hat{\beta}$, and the statistic $t$ follows a chi-squared distribution with one degree of freedom.

**Post-GWAS processing**. The threshold used to determine significant SNPs was calculated by dividing the standard genome-wide significant threshold of $5 \times 10^{-8}$ by the number of effective tests accounting for all variants tested and all metabolites. To estimate the number of effective tests, we first did a PC analysis of our 158 metabolites. Then, we calculated the number of PCs that explained 99% of the total variance. We obtained 39 effective tests. The significance threshold was then $1.28 \times 10^{-9}$.

Because of the great number of signals, we chose to summarize our results by genomic regions, corresponding to approximately independent LD blocks. We sliced the genome in 1703 independent regions based on a recombination map recently described by Berisa et al.[23]. These regions are 10 kb to 26 Mb long, with an average size of 1.6 Mb. For each region, we kept the SNP with the best p-value obtained by either STD or CMS. We then used the UCSC database to assign the closest gene to each SNP, with a maximum distance of 100 kb.

**GWAS of delta between baseline and follow-up**. We used data from baseline and follow-up studies to perform GWAS of the difference between the two time points, divided by the age difference. We called that variable $\Delta_{fb}$:

$$\Delta_{fb} = \frac{f - b}{age_f - age_b} \tag{3}$$

where $f$ and $b$ are metabolite measurements at follow-up and baseline, respectively. As for baseline data analysis, we used STD and CMS approaches, with covariates pre-selection based on AIC. Confounding factors used for the baseline analysis were also included as covariate in all $\Delta_{bf}$ analyses. We did not adjust for baseline value in the main analysis.

**Interaction analyses**. We performed two follow-up interaction analyses for subset of SNP-metabolite associations identified in the GWAS. First, we assessed SNP-by-age interaction effect in both baseline and follow-up analyses for the subset of SNP showing significant effects on $\Delta_{bf}$ in metabolite levels between baseline and follow-up ($\Delta_{bf}$). In practice, we applied a standard linear regression between the corresponding outcome and genetic variant, adjusting for the same potential confounding factors as in the primary GWAS analysis, and adding the interaction term $\beta_{int} G * age$:

$$Y \sim \beta_G G + \boldsymbol{\beta_c} \mathbf{C} + \beta_{age} age + \beta_{int} G * age \tag{4}$$

Second, we assessed potential SNP-by-statin interaction for the 588 regions identified in the primary GWAS analysis. In that specific analysis, we included all statin users (which were removed in the primary analysis for some metabolites, as explained before) and performed linear regression between each metabolite and the best SNP in the associated region (minimum $p$ value). The regression was adjusted by confounding factors and included the interaction term $\beta_{int} G * statin$:

$$Y \sim \beta_G G + \boldsymbol{\beta_c} \mathbf{C} + \beta_{statin} statin + \beta_{int} G * statin \tag{5}$$

**Trend test**. To assess the significance of enrichment for positive or negative effects observed for the identified core regulator genes, we implemented a multivariate test

of all associations that accounted for the correlation between metabolites analyzed jointly. For each gene, we selected the top associated variants across all metabolites (as defined in Table 1), extracted the corresponding single metabolite $z$ score statistics to form a vector $\mathbf{z} = (z_1, z_2, \ldots, z_k)$ where $k$ is the number of metabolites analyzed. We then derived the following multivariate statistics:

$$T = \frac{\left(\sum_{i=1\ldots k} z_i\right)^2}{1^t \Omega 1} \tag{6}$$

where 1 is a $1 \times k$ vector of 1, and $\Omega$ is the phenotypic correlation between the $k$ metabolites analyzed jointly. Under the null hypothesis of no association between the variant tested and any of the $k$ metabolites, $T$ follows a central chi-square distribution with $k$ degree of freedom.

**Heritability.** We first took a set of 3342 individuals corresponding to the intersection between baseline and follow-up data. The baseline and follow-up phenotypes were combined, normalized, and separated into baseline and follow-up series, so the normalized phenotypes at baseline and follow-up were directly comparable (i.e., equal normalized phenotypes at baseline and follow-up correspond to equal raw phenotypes). We used GCTA's bivariate REML[63] and included 10 genetic PCs, age, and age[2] as fixed effects. The effect sizes of the aformentioned fixed effects were strongly correlated at each time point ($\rho > 0.6$) and there were minimal differences in variance explained ($< 5\%$). Heritability estimates at the two time points were plotted using *circlize* R package[64], whereas the complete GCTA output, including genetic and environmental variance estimates, genetic and environmental covariances, and LRT $p$ values for genetic correlation are provided in Supplementary Data 13.

**Reporting summary.** Further information on research design is available in the Nature Research Reporting Summary linked to this article.

## Data availability
Complete summary statistics for both the standard test and CMS, along meta-information are available at http://statgen.pasteur.fr/Download.html. Standard GWAS summary results are also available on the NHGRI-EBI Catalog of published GWAS. All other data are contained in the article and its supplementary information or available upon reasonable request.

## Code availability
The GWAS association screening was performed using the CMS approach. The code developed is available at: https://gitlab.pasteur.fr/statistical-genetics/runCMS/. The bivariate heritability analysis was performed using GCTA's bivariate REML https://cnsgenomics.com/software/gcta/. Plotting of the gene-metabolites network was done using Cytoscape: https://cytoscape.org/. All others analyses and plots were done using the R software: https://www.r-project.org/.

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

## Acknowledgements

We thank the METSIM individuals who participated in this study. This study was funded by National Institutes of Health (NIH) grants R03DE025665, R21HG007687, HL-095056, HL-28481, and U01 DK105561.

## Author contributions

A.G., J.M., and H.A. performed all individual-level data analyses. A.G. and H.A. drafted the manuscript. H.A., A.V., and H.J. performed analyses based on summary level data. N.Z., P.P., H.A., and A.K. developed the analytical plan and supervised the work. A.K., M.A.-K., and M.L provided expertize on metabolites analysis. All authors contributed revisions to the manuscript.

## Competing interests

The authors declare no competing interests.
