## [Peer Review File · Nature Communications]

Reviewers' Comments:

Reviewer #1:

Remarks to the Author:

The authors have performed a genome-wide association study for 158 serum metabolites in 6,263 participants of the Finnish METSIM Study. To improve the statistical power for identifying SNP-metabolite associations, they utilize a multivariate approach that accounts for the high correlation structure between the metabolites. They find 588 genome-wide significant SNP-metabolite associations of which 248 are reported to be novel. The authors also identify genetic associations with changes in metabolite levels across two measurement time points, 5 years apart, and find evidence of genetic interactions with statin intake. The study contains novel findings and implements methodological approaches that help to look into the data from an integrated perspective. I have the following comments and questions to the authors:

1. The authors find that 13 genes capture over 75% of all associations. However, the majority of the studied metabolites (62%) are lipoproteins which are often strongly correlated, and nearly all metabolites are lipid-related. It may be worth noting that future analyses looking at a wider range of metabolites would likely lead to different findings regarding the "master regulator" genes.
2. In the replication analyses, 71.6% of the associations replicated at a nominal threshold of 5%, which seems somewhat low, but might be just due to limited power. Nevertheless, it would be helpful to plot the effect sizes from the discovery and replication studies for each of these variants, to visualize how consistent the effect sizes were between the studies.
3. The authors restricted the annotation to a certain gene by a distance threshold of 100 kb. However, the causal gene may often be located much further away from the variant. Thus, I suggest the authors do not set a distance limit, but simply annotate the SNPs according to the nearest gene.
4. It seems the present study did not include imputation of the genome-wide variant data. What is the reason for this?
5. In the analyses for changes in metabolite levels, not adjusting for the baseline value leads to some level of bias because of regression to the mean. For the SNPs significantly associated with changes in metabolites, it would be reassuring to show that the associations are also significant when adjusting for the baseline value of the metabolite.
6. In Figure 4, it seems that a) and b) should be swapped in the Figure legend?

Reviewer #2:

Remarks to the Author:

Overall:

Gallois et al perform a genetic association study of ~665K genome-wide array genotypes with 158 circulating metabolites, most related to lipoproteins. They both use a univariate and multivariate framework. Through this approach, they detect several SNP-metabolite associations, evaluate the influence from statin use, and evaluate the influence from aging. When describing novel findings as well as novelty of their method, the authors do not include several relevant papers highlighted below that have done similar analyses with similar methods. Highlighting this is the lack of discussing the distinguishing factors of this paper versus a recent metabolomics GWAS in this cohort, which presumably is inclusive of all of these samples (Teslovich T et al. Human Molecular Genetics. 2018). Additional concerns are described below:

Major:

1. Recently, a GWAS of metabolomics in METSIM was published (Teslovich T et al. Human Molecular Genetics. 2018). Surprisingly, this paper did not cite that effort or compare/contrast findings. This likely influences novelty of many of the findings.
2. The authors describe that a key novel aspect of their approach is that they leverage the correlation structure of metabolites for discovery, unlike prior approaches. However, there are other projects that have used correlation structure and network construction of metabolites to similar do genetic association analyses (Inouye M et al. PLoS Genet. 2012; Krumsiek J et al. PLoS Genet. 2012; Nath A et al Genome Biology. 2017; etc). Since such network-based approaches previously have been used for genetic association analyses of metabolomics, please compare these with the current study.
3. When interrogating for known versus novel findings, better datasets for lipid-related GWAS are (versus Teslovich T et al. Nature. 2009) Willer CJ et al Nature Genetics 2013, Surakka I et al. Nature Genetics 2015, Klarin D et al Nature Genetics 2018, and Natarajan P et al. Nature Communications. 2018. Interrogation of these various datasets will change the number of novel loci detected. There are also several other metabolomics GWAS that should be included in interrogation but are not here – Rhee EP et al. Cell Metab 2013, Inouye M et al PLoS Genetics. 2012, Mozaffarian D et al. American Journal of Clinical Nutrition 2014, Rhee EP et al Nature Communications 2016, Yu B et al Sci Adv 2016, Tabassum R et al bioRxiv 2018.
4. The authors highlight the pleiotropy of metabolite-associated variants with multiple metabolites. It currently comes across as descriptive with little known insight. It is already appreciated that many lipid-associated variants are associated with multiple major lipid fractions. With more refined phenotyping, what does the pleiotropy tell us (beyond the fact that it exists)? For example, are there variants at the same locus (particularly at the site of “master metabolic regulator genes” in lines 103-104) that have different metabolite associations? If this is the cause, what mechanistic insight can we glean?
5. When the authors interrogated associated variants with other associations in the literature, they may have used PhenoScanner or the GWAS catalog. For example, the PhenoScanner results for rs10468017, one of the LIPC loci associated variants, included “cardiovascular disease risk factors” in PMID 21943158. The authors, in the text, say that the variant is associated with “cardiovascular disease.” However, upon review of the paper, the association is with “HDL cholesterol” and not any cardiovascular disease. Additionally, “metabolic syndrome” is another result of PhenoScanner which is incorporated in the text; however, the paper (PMID 21386085) shows association of this variant for bivariate phenotypes HDL-C/glucose and waist circumference/HDL-C, not metabolic syndrome. This inflates the disease-relevance of this locus. I would encourage authors to interrogate the associations for cited papers and only state the prior association described in the cited papers, and not rely on publicly available databases.
6. In the interaction analyses, with the analysis of largely common non-coding variants but without the analysis of disruptive coding variants or gene expression data, how are there claims about effects on genes? For example, the strongest statin interaction association implicates APOC1 in these analyses. It’s not clear to me how APOC1 is distinguished. This locus has multiple other relevant apolipoproteins – APOE, APOC2, and APOC4. APOE has a long well-studied history with LDL cholesterol.
7. The discussion is under-developed. The authors start the discussion with stating: “Metabolites have been implicated as important factors in many human diseases. Deciphering the role of genetics in controlling circulating metabolites and its dependence on clinical and environmental characteristics is one of the many challenges facing the human genetics community.” However, the discussion does not describe how their genetic analyses improve our understanding of metabolite-disease relationships.

Minor:

1. Line 36: “a biomarkers” should be “biomarkers.” But, are biomarkers assessed because they are influenced by various exposures? Aren’t they assessed because they have prognostic and potential therapeutic implications?
2. Line 66: “98 lipoproteins” should be “98 lipoprotein components”
3. Line 86: italicize “in silico”

4. Please italicize gene names throughout.
5. The Methods section should include an overview of the metabolite profiling approach.
6. In conventional gene-discovery analyses of lipid-related traits, sex is included as a covariate. Why wasn't it included here?
7. How are genes identified within the Figure 2 schematic? I suspect this is based on the nearest mapped gene to the strongest associated SNP. This may not be the best way to identify the causal gene. For example, one gene highlighted is CELSR2 but functional data implicates SORT1 as the causal gene at this locus (Musunuru K et al. Nature 2010).
8. What is meant by "LIPC expression" in line 126? I would suggest an alternative term since gene expression is not included in this study.
9. Different statins and doses are likely to have differential metabolite influences. How was this considered in the interaction analyses?
10. The text indicates that Figure 4a is for statin interaction and Figure 4b is for age interaction. The Figure 4 legend has the reverse. Please clarify.
11. In addition to comparing heritabilities at baseline and follow-up, heritabilities for age ranges would be helpful.
12. In the Discussion limitations section: "First, while the CMS approach was very successful in our data, doubling the number of identified associations, the current implementation cannot handle related individuals in reasonable computational time. It means that, pending additional development, the power of future studies can be further improved by adding the related individuals in the model." These descriptions are fairly qualitative and unclear – it would be helpful to be more granular.

Reviewer #3:

Remarks to the Author:

The manuscript describes a GWAS of >150 blood metabolite measures in the METSIM cohort. The analysis includes both a "traditional" single variant GWAS and a CMS method intended to increase power by taking into account correlation between multiple phenotypes analyzed in a given study. The authors identify hundreds of associated variants across the hundreds of phenotypes analyzed. The authors suggest that many of these associations are novel and primarily the product of the novel analytical methods employed. Additionally, the authors test for genetic associations with change in metabolite measures over time and for SNP-age and SNP-statin interaction effects.

Comments:

1) The authors have clearly not undertaken a thorough assessment of published literature for blood metabolite levels. Hundreds of associations have previously been published for these phenotypes from the SAME COHORT data in larger sample sizes (see Davis J, et al. PLoS Genetics, 2017 and Teslovich T, et al. Hum. Mol. Genet., 2018). Yet the papers are not even mentioned, nor accounted for when the authors discuss novelty. This would give a much more accurate assessment of the value of the CMS method and other analyses included in the current study.

2) Along the same lines, while the authors provide summary counts of novel loci and loci discovered only by the CMS method, there is little discussion of the properties of these discoveries from either the biological or statistical perspective.

a) I appreciate that potentially hundreds of novel loci are hard to characterize, but there must be some insights to be gleaned from these new results (otherwise, what is the point?).

b) I also think there is substantial value in better characterizing the value of the CMS method discoveries. I'm not entirely sure what the best way(s) to look at this are, but a couple of possible questions: What is the average power gain/decrease in variance compared to the "standard" model? Are certain types of relationships better powered for discovery than others? Are these "real world" discoveries consistent with expectation? What is the per trait (and average) increase in variance explained? How important and easy is it to determine what model produces the strongest association? Is the increase in power worth the loss of related individuals?

3) There is a lack of clarity in the results section between the terms "signal," "locus," and "SNP." Please clarify how each of these terms are used, mention the definition of a locus (particularly as it is somewhat unorthodox), and how associations are determined and accounted across traits. Is it 248 SNPs that haven't been associated before or 248 loci?

4) 71% replication rate suggests a high false positive rate within the novel/CMS associations. Does this suggest that the method has poor type 1 error, that many of these new associations are the result of winner's curse and thus underpowered for replication, or is there another possible reason for the high rate of variants that do not replicate? (see also #7)

5) Why wasn't imputation performed?

6) In the age effect analysis, more could be made of the 4 associations that are not significant in baseline analysis, whether these are truly "age-dependent" associations or statistical quirks, and what this might mean for underlying biology.

7) In the absence of the now standard mixed model approach, I see no assessment or correction for potential population stratification. Were PCs calculated, incorporated into the association models?

8) Figure 4 A and B appear to be swapped vis a vis the descriptions in the descriptions. Also, why are different numbers of bars present for each trait? What was the threshold/criteria for including a trait for a given gene? Is the same SNP used for each trait for each gene?

9) the use of "mutation" in the introduction seems inappropriate.

10) check usage of locus vs. loci

Editor

We would specifically ask that you put your findings in context with the published literature and provide a fair assessment of the novelty of the identified SNPs/loci.

Please be aware that for certain types of new data, including most types of genetic data (i.e. GWAS summary statistics in this case), journal policy is that deposition in a community-endorsed, public repository is generally mandatory prior to publication. Please include a statement about data availability in your point-by-point letter accompanying your revisions.

Reviewer #1 (Remarks to the Author):

The authors have performed a genome-wide association study for 158 serum metabolites in 6,263 participants of the Finnish METSIM Study. To improve the statistical power for identifying SNP-metabolite associations, they utilize a multivariate approach that accounts for the high correlation structure between the metabolites. They find 588 genome-wide significant SNP-metabolite associations of which 248 are reported to be novel. The authors also identify genetic associations with changes in metabolite levels across two measurement time points, 5 years apart, and find evidence of genetic interactions with statin intake. The study contains novel findings and implements methodological approaches that help to look into the data from an integrated perspective. I have the following comments and questions to the authors:

1. The authors find that 13 genes capture over 75% of all associations. However, the majority of the studied metabolites (62%) are lipoproteins which are often strongly correlated, and nearly all metabolites are lipid-related. It may be worth noting that future analyses looking at a wider range of metabolites would likely lead to different findings regarding the “master regulator” genes.

We highlighted that these results mostly address Lipid metabolites, and that our analyses do not preclude similar results for other metabolites. In particular we now added in the discussion that:

“Fourth, our metabolite panel included mostly serum lipids, and the presence of a limited number of master regulators only apply to those phenotypes. Whether other sets of related metabolites have a similar genetic architecture, or even share the same regulators, would have to be determine in cohort with a broader range of metabolites.”

2. In the replication analyses, 71.6% of the associations replicated at a nominal threshold of 5%, which seems somewhat low, but might be just due to limited power. Nevertheless, it would be helpful to plot the effect sizes from the discovery and replication studies for each of these variants, to visualize how consistent the effect sizes were between the studies.

We now added in **Supplementary Table 3** the variance explained by SNP for each association. We also added a new supplementary figure (**Supplementary Figure 4**) showing the squared correlation at discovery as a function of the $-\log_{10}(p\text{-value})$ at replication using the Kettunen et al 2016 data, which has by far the largest overlap with our study. The figure showed a strong consistency between

the two studies, with a correlation of 0.64 between the two parameters. We used $-\log_{10}(P\text{-value})$ to illustrate in the same figure the performance of the replication in regards of the effect size. On that latter point, the figure clearly shows that low replication was directly proportional to the effect size in METSIM. The variants not replicated at the 0.05 significance threshold have the smallest effect size.

In regards of other comments, we also checked whether the SNPs involved in our new associations were previously identified in the large scale GWAS of total HDL, LDL, TC and TG by Willett et al. We found that 76 out of 86 SNPs (87%) have been previously identified at nominal significance with one of these four phenotypes. (**Supplementary Tables 3 and 6**)

3. The authors restricted the annotation to a certain gene by a distance threshold of 100 kb. However, the causal gene may often be located much further away from the variant. Thus, I suggest the authors do not set a distance limit, but simply annotate the SNPs according to the nearest gene.

We now report the closest gene in all tables.

4. It seems the present study did not include imputation of the genome-wide variant data. What is the reason for this?

The reviewer is correct, we did not use imputed SNPs. The main reason is the prohibitive computational cost of the current CMS implementation. Each CMS analysis of the 150 metabolites on the 600,000 genotypes took a few weeks using over 1,000 cores on the UCLA cluster. We are actively working on a new, improved and much faster implementation of CMS, which we hope, will allow for the analysis of larger number of SNPs and larger sample size in the near future.

5. In the analyses for changes in metabolite levels, not adjusting for the baseline value leads to some level of bias because of regression to the mean. For the SNPs significantly associated with changes in metabolites, it would be reassuring to show that the associations are also significant when adjusting for the baseline value of the metabolite.

We understand the reviewer comments, and we actually asked ourselves the same question of adjusting for baseline or not. Our final decision of not adjusting was motivated by the potential risk of introducing a bias in the estimation of the genetic effect on the slope. Consider the toy modelling example where Y_1 and Y_2 are measurements at baseline and follow-up, respectively: $Y_1 = \beta_1 * G + \epsilon$ and $Y_2 = \beta_2 * G + \epsilon$

The unadjusted slope equals: $D_{\text{unadj}} = Y_2 - Y_1 = (\beta_2 - \beta_1) * G + \epsilon$.

After adjusting for the baseline, and assuming the correlation between Delta and the baseline equals alpha, we have: $D_{\text{adj}} = D - \alpha * Y_1 = (\beta_2 - \beta_1 * (1 + \alpha)) * G + \epsilon$

This is of course an oversimplified model, but from this simple example, one can clearly see that if the genetic effects are identical between Y_1 and Y_2 , there will be no effect on the slope in the unadjusted model. Conversely, if $\beta_1 = \beta_2$, the adjusted model might display a genetic effect estimate with a mean equal to alpha.

Nevertheless, now present in the **Supplementary table 14** the p-value from the adjusted for all top variants. All association remain nominally significant except one. Overall, the significance decreases, although it increases for some variants.

6. In Figure 4, it seems that a) and b) should be swapped in the Figure legend?

Thank you for catching this typo. We have corrected the legend

#####Reviewer #2 (Remarks to the Author):

Overall: Gallois et al perform a genetic association study of ~665K genome-wide array genotypes with 158 circulating metabolites, most related to lipoproteins. They both use a univariate and multivariate framework. Through this approach, they detect several SNP-metabolite associations, evaluate the influence from statin use, and evaluate the influence from aging. When describing novel findings as well as novelty of their method, the authors do not include several relevant papers highlighted below that have done similar analyses with similar methods. Highlighting this is the lack of discussing the distinguishing factors of this paper versus a recent metabolomics GWAS in this cohort, which presumably is inclusive of all of these samples (Teslovich T et al. Human Molecular Genetics. 2018). Additional concerns are described below:

1. Recently, a GWAS of metabolomics in METSIM was published (Teslovich T et al. Human Molecular Genetics. 2018). Surprisingly, this paper did not cite that effort or compare/contrast findings. This likely influences novelty of many of the findings.

We thank the reviewer for pointing this paper, that was not published at the time we started the replication analysis, and we obviously missed it in meantime. We have now included it in our count for new hits and replication. Nevertheless, we note that this paper only focused on nine amino acids, while our analysis included 158 metabolites. In practice, it did not change our total count – all reported association from our manuscript that overlap with Teslovitch et al 2018, were also reported by Kettunen et al 2016, and were therefore not counted as new association in our first submission.

More generally, in response to comments from all reviewers regarding the comparison with previous GWAS, we report at the end of our response an updated list of all additional studies we now incorporate, along their characteristics. Overall, the addition of these studies did not change our count of new associations (we found only one additional known association, bringing the total of new association to 247), as all additional locus-metabolites studies were already reported by either Kettunen et al, 2016, Rhee et al 2013, or Teslovitch et al, 2010. Note that we removed the latter reference and now use Klarin et al 2018 and Willer et al 2013 for comparison with total lipids.

2. The authors describe that a key novel aspect of their approach is that they leverage the correlation structure of metabolites for discovery, unlike prior approaches. However, there are other projects that have used correlation structure and network construction of metabolites to similar do genetic association analyses (Inouye M et al. PLoS Genet. 2012; Krumsiek J et al. PLoS Genet. 2012; Nath A et al Genome Biology. 2017; etc). Since such network-based approaches

previously have been used for genetic association analyses of metabolomics, please compare these with the current study.

We thank the reviewer for pointing to these alternative methods. We would like to highlight that the goal here was not to demonstrate the performance of CMS as compared to other existing methods, which has been done in a previous study (PMID= 29038595).

Moreover, the CMS approach is a covariate adjustment strategy, and our application consists in a standard univariate phenotype-genotype association screening. It is similar to any standard univariate GWAS, and provides univariate regression coefficient and the corresponding p -value. The only difference is the boost in power achieved thanks to the covariate adjustment informed by the CMS algorithm. The papers mentioned by the reviewer describe interesting approaches but they have fundamentally different objectives. Nevertheless, we now discuss briefly the various strategies possible for analyzing multi-trait data, and cite the above references in the discussion section.

Inouye M et al. PLoS Genet. 2012 – this study performed multivariate tests of association of multiple metabolites. Such tests are based on a composite null hypothesis (i.e. the tested variant is associated with at least one of the metabolites tested jointly). This can have increased power as compared to univariate test, but this is at the cost of reduced granularity, as a positive association does not say which metabolite is actually associated with the genotype tested.

Krumsiek J et al. PLoS Genet. 2012 – this study used Gaussian graphical modelling to predict the biochemical identities of unknown metabolites. We believe this is extremely distinct in objective from the study we performed.

Nath A et al Genome Biology. 2017 - this study is a complex integrative analysis merging metabolite, transcriptomic and genetic data, as well as annotation data to infer potential multilevel networks, focusing on immunity-related pathways. Again, while we appreciate the great work performed in this study, we believe the objective of this study is far outside the scope of our work.

3. When interrogating for known versus novel findings, better datasets for lipid-related GWAS are (versus Teslovich T et al. Nature. 2009) Willer CJ et al Nature Genetics 2013, Surakka I et al. Nature Genetics 2015, Klarin D et al Nature Genetics 2018, and Natarajan P et al. Nature Communications. 2018. Interrogation of these various datasets will change the number of novel loci detected. There are also several other metabolomics GWAS that should be included in interrogation but are not here – Rhee EP et al. Cell Metab 2013, Inouye M et al PLoS Genetics. 2012, Mozaffarian D et al. American Journal of Clinical Nutrition 2014, Rhee EP et al Nature Communications 2016, Yu B et al Sci Adv 2016, Tabassum R et al bioRxiv 2018.

We thank the reviewer for suggesting these papers. We were aware of some of them, but we did not include them because of small sample size, differences in the approaches, and more importantly, a limited number of metabolites studied. As mentioned in the manuscript, we focused on studies with the largest sample size and enough metabolic overlap, as we believed those would be the most relevant for replication and comparison purposes. Nevertheless, we understand the reviewer concern – also raised by reviewer #2 and we now accounted for all of these studies when counting the number of new metabolite-genotypes associations. Note that despite careful and extensive checking we did not find any of our new associations in these additional studies.

Please also note that several of these papers (Willer CJ et al Nature Genetics 2013, Surakka I et al. Nature Genetics 2015, Klarin D et al Nature Genetics 2018, Natarajan P et al. Nature

Communications. 2018), but also other papers (e.g. van Leeuwen et al Nat Comm 2015, Dumitrescu et al PLoS Genet 2011, Wu et al PLoS Genet 2013, Sabatti et al Nat Genet 2009,...) focused only on the overall LDL, HDL, TC and TG – which represent only 4 out of the 158 outcomes we considered. All associations we reported for these phenotypes were already identified by Teslovicht et al. Nature. 2010. Nevertheless, we now include association between our top SNPs and total lipids (HDL, LDL, TG, and TC) as reported by Willer CJ et al Nature Genetics 2013, for which genome-wide summary statistics were available. Importantly, we also now highlight that most of our new association involve known loci from these total lipids GWAS.

Regarding metabolite GWAS, we summarize below the changes we made:

The study by Rhee EP et al. Cell Metab 2013 was already included in our comparison.

We incorporated the one association found between genes FADS1/2 and cis/trans-18:2 fatty acids identified by Mozaffarian D et al. American Journal of Clinical Nutrition 2014 (analysis of five fatty acids in 8013 individuals)

We included the 2 associations, HAL and histidine, and PAH and phenylalanine from Rhee EP et al Nature Communications 2016 (analysis of 217 metabolites and exome variants in 2,076 participants) that overlapped with our analysis.

We also included the three associations from Davis J, et al. PLoS Genetics, 2017 (suggested by reviewer #2, analysis of 72 lipid and lipoprotein traits in 8,372 individuals) overlapping with our analysis: HIF3A and Phospholipids in small VLDL, PLTP and large HDL particles, and LIPG and Phospholipids in medium HDL.

We did not include results from Inouye M et al PLoS Genetics. 2012 (analysis of 130 metabolites in 6,600 individuals) as they are not directly comparable. As discussed above, this study used a composite null model, *i.e.* it does not provide information about which metabolite in the “network” are associated with the tested genetic variants, while our approach allows for detecting association between a single variant and a single metabolite.

We also did not include the results from Yu et al Sci Adv 2016 in our supplementary tables (analysis of 308 metabolites in 1361 African-Americans, which found four genes associated with five metabolites), as there was no overlap in identified signals with our study.

Finally, we also did not include results from Tabassum R et al bioRxiv 2018 (analysis of 141 lipid species in 2,181 individuals) as this study is only in a repository and results can potentially change substantially until final publication.

4. The authors highlight the pleiotropy of metabolite-associated variants with multiple metabolites. It currently comes across as descriptive with little known insight. It is already appreciated that many lipid-associated variants are associated with multiple major lipid fractions. With more refined phenotyping, what does the pleiotropy tell us (beyond the fact that it exists)? For example, are there variants at the same locus (particularly at the site of “master metabolic regulator genes” in lines 103-104) that have different metabolite associations? If this is the cause, what mechanistic insight can we glean?

While we agree that pleiotropy has previously been shown, identifying the targets of master regulators is an important and non-descriptive step towards understanding mechanism. In this study we increased the number of known targets for most genes. Furthermore, pleiotropy was just one of

several important contributions of this paper, others include the first identification of genetic associations to change in metabolites over time and the importance of multivariate methods in metabolic analyses.

We also used the example of LIPC (the gene associated with the highest number of metabolites) to highlight the potentially complex pattern underlying the master regulator. The fine-mapping results match the reviewer's hypothesis, showing that there are likely multiple variants displaying differential effect on the associated metabolites (**Figure 5** and **supplementary tables 9-11**). However, based on our recent experience in other unrelated projects, the extended and systematic analysis of each of the 13 genes would be a yearlong study by itself, and we believe is out of the scope of our study.

Our main text has been updated in these direction as follows:

*"For pleiotropic genes, we observed heterogeneity in the reported top SNP across metabolites (Figure 5a). For example, the top SNP for APOC1 was the same across all 33 associated metabolites (rs445925). Conversely, there was 9 top SNPs for the 75 metabolites associated with LIPC. Part of this heterogeneity might be explained by LD in these regions, but also by the presence of multiple causal variants affecting different metabolites. To investigate this possibility, we applied the FINEMAP²⁶ algorithm using the example of the latter LIPC region after performing additional genotype imputation in that region (**Supplementary Note** and **Supplementary Table 9**). Our analysis identified 3 distinct association signals with consistently high probabilities of causal effects from 7 SNPs and heterogeneous metabolite association patterns, confirming the likely presence of metabolite-specific variants within this gene (**Figure 5** and **Supplementary Tables 10-11**)."*

And the discussion now includes additional details:

"To better understand the role of the master regulators in the etiology of lipoprotein components, which contributed the vast majority of the reported associations, we performed a series of analyses that highlighted a limited number of association patterns. Overall, three genes LIPC, LDLR, and PCSK9 showed global effects, and were associated with all types and sizes of metabolites. On the other hand, some equally pleiotropic genes (in terms of total association reported), such as APOA5 and GCKR, appear to affect specific types of lipids, and VLDL in particular. Many of the master regulator variants also display different top associated variants depending on the metabolites analyzed. While some of this variability might be due to LD in these regions, our fine mapping of the LIPC region showed that the pleiotropic effect of at least some of these genes is likely due to multiple genetic variants with heterogeneous effects on the associated metabolites.

More systematic fine-mapping of all identified associations is out of the scope of this work as it would require high-density SNPs data including rare variants. Future fine-mapping studies may further improve resolution by leveraging functional annotations^{46,47}. Moreover, the fine-mapping analysis we performed for LIPC demonstrated how multi-trait associations might help identify likely causal variants. However, we used a naïve approach that simply aggregates univariate fine-mapping results. More advanced methodologies integrating all information into a single framework could provide more accurate posterior probabilities on likely causal variants."

5. When the authors interrogated associated variants with other associations in the literature, they may have used PhenoScanner or the GWAS catalog. For example, the PhenoScanner results for rs10468017, one of the LIPC loci associated variants, included "cardiovascular disease risk

factors” in PMID 21943158. The authors, in the text, say that the variant is associated with “cardiovascular disease.” However, upon review of the paper, the association is with “HDL cholesterol” and not any cardiovascular disease. Additionally, “metabolic syndrome” is another result of PhenoScanner which is incorporated in the text; however, the paper (PMID 21386085) shows association of this variant for bivariate phenotypes HDL-C/glucose and waist circumference/HDL-C, not metabolic syndrome. This inflates the disease-relevance of this locus. I would encourage authors to interrogate the associations for cited papers and only state the prior association described in the cited papers, and not rely on publicly available databases.

We thank the reviewer for noting these imprecisions. We have removed those specific references, and checked the remaining ones.

6. In the interaction analyses, with the analysis of largely common non-coding variants but without the analysis of disruptive coding variants or gene expression data, how are there claims about effects on genes? For example, the strongest statin interaction association implicates APOC1 in these analyses. It’s not clear to me how APOC1 is distinguished. This locus has multiple other relevant apolipoproteins – APOE, APOC2, and APOC4. APOE has a long well-studied history with LDL cholesterol.

The reviewer is correct that other genes in the vicinity of the top associated variants might be involved. In response to this comment, and comments from other reviewers, we now make clear that associated SNPs were mapped to the closest gene. We also mention this limitation in the discussion and highlight that the identification of the genes involved will require additional extensive analysis. The added paragraph is as follows:

“Fourth, we mapped associated variants with the closest gene. However, extended fine-mapping analysis using sequencing data, as well as gene expression analysis across multiple tissues might demonstrate a mode of action through other genes.”

7. The discussion is under-developed. The authors start the discussion with stating: “Metabolites have been implicated as important factors in many human diseases. Deciphering the role of genetics in controlling circulating metabolites and its dependence on clinical and environmental characteristics is one of the many challenges facing the human genetics community.” However, the discussion does not describe how their genetic analyses improve our understanding of metabolite-disease relationships.

We performed an overhaul of the discussion section, adding in particular two paragraphs that discuss improving discoveries through new methodologies such as CMS, and the role of master regulators on lipoproteins. We also clarify that our goal was to improve our knowledge on the genetic component of metabolites –because of their link with human diseases. The beginning of the discussion now says:

Metabolites have been implicated as important factors in many human diseases and identifying the genetic variants controlling circulating metabolites and their relationship to clinical and environmental characteristics is one of the many challenges facing the human genetics community.

Minor:

1. Line 36: “a biomarkers” should be “biomarkers.” But, are biomarkers assessed because they are influenced by various exposures? Aren’t they assessed because they have prognostic and potential therapeutic implications?

Done

2. Line 66: “98 lipoproteins” should be “98 lipoprotein components”

Done

3. Line 86: italicize “in silico”

Done

4. Please italicize gene names throughout.

Done

5. The Methods section should include an overview of the metabolite profiling approach.

Done

6. In conventional gene-discovery analyses of lipid-related traits, sex is included as a covariate. Why wasn’t it included here?

The METSIM study includes males only.

7. How are genes identified within the Figure 2 schematic? I suspect this is based on the nearest mapped gene to the strongest associated SNP. This may not be the best way to identify the causal gene. For example, one gene highlighted is CELSR2 but functional data implicates SORT1 as the causal gene at this locus (Musunuru K et al. Nature 2010).

The reviewer is correct. This is based on the nearest mapped gene. It is indeed possible that the effect of the identified goes through another close by gene. Mapping associated variants to the acting gene is a complex task and would be out of the scope of this study. In response to this point and major comment #5, we now highlight this important point in the discussion:

“Fourth, we mapped associated variants with the closest gene. However, extended fine-mapping analysis using sequencing data, as well as gene expression analysis across multiple tissues might demonstrate a mode of action through other genes.”

8. What is meant by “LIPC expression” in line 126? I would suggest an alternative term since gene expression is not included in this study.

We now clarified that association with expression of LIPC was reported in a previous study, and rewrote that paragraph as follows:

“It is located in a region harbouring H3K4me1/H3K4me3 and H3K27ac/H3K9ac marks of promoter and enhancer in adipose derived Mesenchymal Stem Cell Cultured Cells. This variant was also reported to be associated with LIPC expression in human liver tissue in a previous study³⁴, suggesting a potential mode of action through the regulation of LIPC expression.”

9. Different statins and doses are likely to have differential metabolite influences. How was this considered in the interaction analyses?

We didn't have access to the dosage and type. While various dosage and statin type might indeed have differential metabolite influences, we expect a limited impact in our study. Indeed, in the general Finish population, approximately 70% of the statin user are on Simva-Statins.

Assuming similar proportions in our sample, even if the information was available, we would not be powered enough to explore heterogeneity across statin type. Nevertheless, given that we identified an interaction this will be an interesting and important question to explore when larger cohorts are available.

10. The text indicates that Figure 4a is for statin interaction and Figure 4b is for age interaction. The Figure 4 legend has the reverse. Please clarify.

Thank you for catching this typo. We have corrected the legend.

11. In addition to comparing heritabilities at baseline and follow-up, heritabilities for age ranges would be helpful.

We definitely agree this would be an interesting analysis, but there are unfortunately not enough individuals to meaningfully distinguish heritabilities after splitting into age ranges in our data. That is why we used baseline and follow up.

We are aware of at least one study that looked across age ranges for BMI and Height (Robinson et al, 2017, PMID= 28692066), but they had access to a sample size of more than 43,000 individuals. Moreover, while the study identified a potential change of heritability over age range, they considered various alternative advanced models, and highlighted a number of methodological and computational challenges.

12. In the Discussion limitations section: “First, while the CMS approach was very successful in our data, doubling the number of identified associations, the current implementation cannot handle related individuals in reasonable computational time. It means that, pending additional development, the power of future studies can be further improved by adding the related individuals in the model.” These descriptions are fairly qualitative and unclear – it would be helpful to be more granular.

We updated the text to clarify this statement as follows:

“First, the CMS approach is currently limited to the analysis of unrelated individuals, and has a much higher computational cost than standard linear regression. As a result, we had to remove related individuals from our analysis and limit the screening to the 600K genotyped variants. We are currently developing an improved implementation of CMS addressing these limitations. This will allow for further increases in power thanks to the addition of related individuals, and the analysis of imputed genotypes that might help refine signal at associated regions.”

Reviewer #3 (Remarks to the Author):

The manuscript describes a GWAS of >150 blood metabolite measures in the METSIM cohort. The analysis includes both a "traditional" single variant GWAS and a CMS method intended to increase power by taking into account correlation between multiple phenotypes analyzed in a given study. The authors identify hundreds of associated variants across the hundreds of phenotypes analyzed. The authors suggest that many of these associations are novel and primarily the product of the novel analytical methods employed. Additionally, the authors test for genetic associations with change in metabolite measures over time and for SNP-age and SNP-statin interaction effects.

1) The authors have clearly not undertaken a thorough assessment of published literature for blood metabolite levels. Hundreds of associations have previously been published for these phenotypes from the SAME COHORT data in larger sample sizes (see Davis J, et al. PLoS Genetics, 2017 and Teslovich T, et al. Hum. Mol. Genet., 2018). Yet the papers are not even mentioned, nor accounted for when the authors discuss novelty. This would give a much more accurate assessment of the value of the CMS method and other analyses included in the current study.

We thank the reviewer for pointing these two references. Reviewer #1 suggested also additional references. We now extended the number of studies used for comparison purposes. Below is a copy of our response to this previous comment:

Regarding metabolite GWAS, we summarize below the changes we made:

The study by Rhee EP et al. Cell Metab 2013 was already included in our comparison.

We incorporated the one association found between genes FADS1/2 and cis/trans-18:2 fatty acids identified by Mozaffarian D et al. American Journal of Clinical Nutrition 2014 (analysis of five fatty acids in 8013 individuals)

We included the 2 associations, HAL and histidine, and PAH and phenylalanine from Rhee EP et al Nature Communications 2016 (analysis of 217 metabolites and exome variants in 2,076 participants) that overlapped with our analysis.

We also included the three associations from Davis J, et al. PLoS Genetics, 2017 (analysis of 72 lipid and lipoprotein traits in 8,372 individuals) overlapping with our analysis: HIF3A and Phospholipids in small VLDL, PLTP and large HDL particles, and LIPG and Phospholipids in medium HDL.

We did not include results from Inouye M et al PLoS Genetics. 2012 (analysis of 130 metabolites in 6,600 individuals) as they are not directly comparable. This study used a composite null model, *i.e.* it does not provide information about which metabolite in the “network” are associated with the

tested genetic variants, while our approach allows for detecting association between a single variant and a single metabolite.

We also did not include the results from Yu et al Sci Adv 2016 in our supplementary tables (analysis of 308 metabolites in 1361 African-Americans, which found four genes associated with five metabolites), as there was no overlap in identified signal with our study.

Finally, we also did not include results from Tabassum R et al bioRxiv 2018 (analysis of 141 lipid species in 2,181 individuals) as this study is only in a repository, and results can potentially change until final publication.

2) Along the same lines, while the authors provide summary counts of novel loci and loci discovered only by the CMS method, there is little discussion of the properties of these discoveries from either the biological or statistical perspective.

a) I appreciate that potentially hundreds of novel loci are hard to characterize, but there must be some insights to be gleaned from these new results (otherwise, what is the point?).

In response to this comment and to other reviewer comments we now provide a refined description of the association pattern for the 13 “master regulators” identified, along 2 additional figures (**Figure 4** and **Supplementary Figure 6**). The results section now includes the following description:

*“To better understand the role of these master regulators we performed two additional analyses. First, we appreciate that the observed pleiotropy for these genes is relative, because of the strong correlation across phenotypes. To approximate the number of independent components associated with each gene, we derived the number of principal components (PCs) necessary to explain percentages of the total variance of the corresponding associated metabolites (**Supplementary Table 8**). Overall, while there is, as expected, a decrease in the total number of independent components, the number of potential meaningful association remains quite high. It required on average 10%, 30%, and 50% of the PCs to explain 90%, 99%, and 99.9% of the total variance, respectively. There was limited variability across genes, and similar numbers were observed when focusing only on lipoproteins. For example, for LIPC, it required 19 and 12 PCs to explain 99% of the variance of the 75 metabolites, and the 43 lipoproteins, respectively.*

*Second, we synthesized the results across the lipoproteins, which contribute to the majority of the observed associations (**Figure 4**). Overall, the genes show homogeneous association by lipoprotein class (particles, lipids, phospholipids, cholesterol, cholesterol ester, free cholesterol, and triglyceride), some variability by size (extremely large, very large, medium, small, and very small), and strong heterogeneity by type (VLDL, HDL, LDL, and IDL). We observed four major patterns: i) TRIB1, LPL, GCKR, GALNT2, and APOA5 are mostly associated with VLDL of average size; ii) PCSK9, LIPC, and LDLR are associated with most types and with large to very small lipoproteins; iii) CELSR2 and APOC1 are associated with LDL and IDL of average size; and iv), CETP, FADS1-2, and DOCK7 are mostly associated with VLDL and HDL, but with differences in the size of the associated lipoprotein: FADS1-2 and DOCK7 are enriched for association with very large and medium size, respectively, while CETP displays association with lipoproteins of all sizes. Looking at other lipoprotein associated genes, we found that some might fit in these categories, but a large majority appears to have more targeted effects, being associated with specific types and sizes (**Supplementary Figure 6**).”*

And the discussion section also extends the description of these results and potential future fine-mapping:

“To better understand the role of the master regulators in the etiology of lipoprotein components, which contributed the vast majority of the reported associations, we performed a series of analyses that highlighted a limited number of association patterns. Overall, three genes LIPC, LDLR, and PCSK9 showed global effects, and were associated with all types and sizes of metabolites. On the other hand, some equally pleiotropic genes (in terms of total association reported), such as APOA5 and GCKR, appear to affect specific types of lipids, and VLDL in particular. Many of the master regulator variants also display different top associated variants depending on the metabolites analyzed. While some of this variability might be due to LD in these regions, our fine mapping of the LIPC region showed that the pleiotropic effect of at least some of these genes is likely due to multiple genetic variants with heterogeneous effects on the associated metabolites.

More systematic fine-mapping of all identified associations is out of the scope of this work as it would require high-density SNPs data including rare variants. Future fine-mapping studies may further improve resolution by leveraging functional annotations^{50,51}. Moreover, the fine-mapping analysis we performed for LIPC demonstrated how multi-trait associations might help identify likely causal variants. However, we used a naïve approach that simply aggregates univariate fine-mapping results. More advanced methodologies integrating all information into a single framework could provide more accurate posterior probabilities on likely causal variants.”

b) I also think there is substantial value in better characterizing the value of the CMS method discoveries. I'm not entirely sure what the best way(s) to look at this are, but a couple of possible questions: What is the average power gain/decrease in variance compared to the "standard" model? Are certain types of relationships better powered for discovery than others? Are these "real world" discoveries consistent with expectation? What is the per trait (and average) increase in variance explained? How important and easy is it to determine what model produces the strongest association? Is the increase in power worth the loss of related individuals?

This study is the first large-scale application of CMS, and the reviewer is right that additional characterization of the advantage of the method would be of interest. We now added a new paragraph in the results and a new figure in the main text (**Figure 2**). Overall, we highlight that that CMS increases the detection of variant with low effect size, substantially boosting power

Panel **c** of that new figure also addresses the cost/benefit of using CMS against including the approximately 3,000 related individuals. It shows the gain in power expressed as the increase in sample size –referred in the text as N_{eff} , the effective sample size achieved thanks to the adjustment for covariates selected by CMS. For the identified association the average N_{eff} was 14,000, which corresponds to a 2.2 fold increase as compared to the baseline sample size of 6,263 individuals.

3) There is a lack of clarity in the results section between the terms "signal," "locus," and "SNP." Please clarify how each of these terms are used, mention the definition of a locus (particularly as it is somewhat unorthodox), and how associations are determined and accounted across traits. Is it 248 SNPs that haven't been associated before or 248 loci?

We now describe associations in terms of SNP or “region”, and highlight in the text that: *“To approximate the number of independent associations identified, we grouped significant SNPs in independent linkage disequilibrium (LD) blocks²¹, denoted further as regions (**Online Methods**)”*

4) 71% replication rate suggests a high false positive rate within the novel/CMS associations. Does this suggest that the method has poor type 1 error, that many of these new associations are the result of winner's curse and thus underpowered for replication, or is there another possible reason for the high rate of variants that do not replicate? (see also #7)

To determine the most likely source of non-replication, we plotted in **supplementary Figure 4** the 442 region-metabolite associations with data for both the discovery and replication stage (from Kettunen et al, 2016), showing the variance explained by the top variant (defined as R^2 , the squared correlation) in METSIM, as a function of the $-\log_{10}(P\text{-value})$ observed in the Kettunen et al. study for the same variant. We observed a strong correlation of 0.63. The average R^2 in METSIM was 0.0023 and 0.0068 for the associations not nominally significant and nominally significant in Kettunen et al 2016, respectively. Thus, the data show evidence for a lack of power in the replication instead of a poor type I error rate.

As an additional validation, we also now compared our results with summary statistics for serum HDL, LDL, TC and TG from Willer et al 2013 (**Supplementary Table 6**). In this comparison approximately 93% of the SNPs showed nominal significance with either of the four phenotypes, which we believe, further confirms the relevance of our findings.

5) Why wasn't imputation performed?

The main reason is the prohibitive computational cost of the current CMS implementation. Each CMS analysis on the 600,000 genotypes took a few weeks. We are actively working on a new, improved and much faster implementation of CMS, which we hope, will allow for the analysis of larger number of SNPs and larger sample size.

6) In the age effect analysis, more could be made of the 4 associations that are not significant in baseline analysis, whether these are truly "age-dependent" associations or statistical quirks, and what this might mean for underlying biology.

There can be much greater power to discover an interaction when subtracting time points because the shared noise between each time point will be removed. Our discovery suggests that a larger sample will reveal a baseline association and therefore this falls into the category of "statistical quirk".

7) In the absence of the now standard mixed model approach, I see no assessment or correction for potential population stratification. Were PCs calculated, incorporated into the association models?

We did not, because all individuals come from a very homogenous population.

8) Figure 4 A and B appear to be swapped vis a vis the descriptions in the descriptions. Also, why are different numbers of bars present for each trait? What was the threshold/criteria for including a trait for a given gene? Is the same SNP used for each trait for each gene?

Thank you for catching this typo. We have corrected the legend

9) the use of "mutation" in the introduction seems inappropriate.

We replaced mutation by "genetic variants"

10) check usage of locus vs. loci

We now replaced the loci/locus terminology by "region"

Reviewers' Comments:

Reviewer #1:

Remarks to the Author:

The authors have addressed my comments satisfactorily - I have no further remarks.

Reviewer #2:

Remarks to the Author:

Overall:

Gallois A et al now provide a substantially revised manuscript describing the relationship of genetic variants of metabolites. The authors provide several new analyses that enrich their paper. The improved incorporation of prior discovery and methodologic papers increases confidence in many claims. I do still have several reservations: gene-level claims remain unjustified, almost all of the "new" associations are for lipid-related traits which are described in larger plasma lipids GWA studies, and biological are still limited but could be improved. These issues are further described below:

Major:

1. The authors have previously described this method (PMID: 29038595). Virtually all "new" discoveries are for lipid-related components (e.g., IDL cholesterol IDL triglycerides, etc). As expected, these are highly correlated with conventional plasma lipid measures in much larger GWAS and so nearly all of these loci have been previously associated with plasma lipids. Both of these features generally diminish novelty. However, per my suggestions and author reviewers, the authors now incorporate additional analyses regarding the identification of so-called "master regulators." I found these analyses quite interesting and appreciate their incorporation. I wasn't able to find a description of identification of major lipid-related patterns. I'd like to better understand how these distinctions were derived. For example, PCSK9 and LDLR are both similarly associated with LDL cholesterol and coronary artery disease, but LIPC is incorporated into this group – this gene is associated with HDL cholesterol and not with coronary artery disease. As such, whatever criteria linking these genes together is unlikely to be biologically important.

2. The authors now incorporate fine-mapping of LIPC for the top metabolites. Presumably, these variants were all previously associated with plasma lipids in larger datasets. Therefore, wouldn't fine mapping of a larger dataset using conventional lipids be more powerful to identify causal variants at LIPC? Do the heterogeneous SNP-metabolite associations for the putative causal SNPs provide any new insight?

3. In the prior version, I described the issue with attributing genes to SNPs by proximity, particularly for lipids. A key challenge is the APOE-C1-C2-C4 region. The authors addressed this by clarifying that SNP-gene attributions are by nearest gene. Unfortunately, this still doesn't address this major issue, particularly at this locus. The claims made throughout this study are gene-based. The authors highlight APOC1 at this locus but there are coding SNPs in APOE even more strongly associated with lipids (PMID: 30140000). Also, they attribute the 1p13.3 association to CELSR2 but functional studies attribute the SNP associations to another gene that is not the most proximal gene, SORT1 (PMID: 20686566). Also, they attribute 1p31.3 gene to DOCK7 but there is stronger functional and genetic support for ANGPTL3 (PMIDs: 28538136, 28385496). The current analyses do not permit causal gene claims throughout the abstract, results, and discussion, and the authors may be inadvertently highlighting non-causal genes.

4. The statin interaction analyses allow for unique analyses in METSIM, which is a strength of this study. Since statins strongly influence LDL-C, and particularly non-HDL-C, a helpful secondary analysis would be to adjust for LDL-C change, or non-HDL-C change. In general, key unique

aspects of the paper are the statin and age interaction analyses. I would incorporate more of a discussion about these analyses than on power improvements from CMS since the CMS method has already been previously described.

Minor:

1. I'm not sure if there are formatting issues but Supplementary Table 6 seems to have some issues. The column labeled "Kettunen 2016" has various p-values which are copied over to "minimum P" after a dark line but there should be another dark line to the right so it's clear that it doesn't apply to Willer 2013. Also, why is the column for "Willer 2013" empty but there are various p-values corresponding to these SNPs for the different lipid fractions in columns U-X?

Reviewer #3:

Remarks to the Author:

On first assessment of the modified manuscript, the authors appear to have worked hard to put their work into proper context with the current literature and address fundamental analytical interpretations of their novel method. Their assessment of major regulators of lipid particle biology perhaps puts into better detail a good deal of known lipid biology, however they provide little insight into the novel associations.

These additional analyses do a reasonable job of attempting to show the variability in discovery is due largely to either power deficits (failure to replicate lower effect signals) and or large power gains (for CMS-only associations). This conclusion is rather bolstered by reasonably good comparison to associations in broader lipid measures studies in much larger sample sizes. I greatly appreciate the deeper insight into the variants discovered by CMS.

I remain concerned that the authors have done an adequate comparison of the literature and publicly available data. For examples, they only list 5 overlapping variants from Davis et al., yet a cursory examination of that publicly available data (<http://csg.sph.umich.edu/boehnke/public/metsim-2017-lipoproteins/>) yields more than 200 overlapping genome-wide significant variants. It was not clear from the tables which of the 247 associations they are claiming to be "novel," but I would be shocked if a good deal of them were not already observed in this earlier (larger) analysis the same data.

Perhaps of equal or more value than the loci these two very similar studies both discovered, is an understanding of the discrepancies between the two. Variability in discoveries from very different study samples is not a major surprise. For both population, cohort, methodological, and chance reasons, loci will differ. However, since these two studies use essentially the same underlying data, it is rather more disturbing not to have more consistency. If there really are 247 novel associations among the 588 presented here, why is the overlap so poor? Similarly, there are dozens (hundreds?) of associations in Davis not observed here. A better understanding of these discrepancies (and a rational explanation for them) would also likely give a better true understanding of the value of the CMS method.

Numerous papers have shown that even seemingly homogeneous populations such as Finland can exhibit geographical structure. While METSIM is from a small region, I do not think it is appropriate not to do any adjustment for population structure. Some adjustment or justification for not adjusting must be provided. This may explain some of the discrepancies discussed above.

Reviewer #1#####

The authors have addressed my comments satisfactorily - I have no further remarks.

Reviewer #2 #####

Overall: Gallois A et al now provide a substantially revised manuscript describing the relationship of genetic variants of metabolites. The authors provide several new analyses that enrich their paper. The improved incorporation of prior discovery and methodologic papers increases confidence in many claims. I do still have several reservations: gene-level claims remain unjustified, almost all of the “new” associations are for lipid-related traits which are described in larger plasma lipids GWA studies, and biological are still limited but could be improved. These issues are further described below.

We thank the reviewer for the encouraging comment.

Major 1. The authors have previously described this method (PMID: 29038595). Virtually all “new” discoveries are for lipid-related components (e.g., IDL cholesterol IDL triglycerides, etc). As expected, these are highly correlated with conventional plasma lipid measures in much larger GWAS and so nearly all of these loci have been previously associated with plasma lipids. Both of these features generally diminish novelty. However, per my suggestions and author reviewers, the authors now incorporate additional analyses regarding the identification of so-called “master regulators.” I found these analyses quite interesting and appreciate their incorporation. I wasn’t able to find a description of identification of major lipid-related patterns. I’d like to better understand how these distinctions were derived. For example, PCSK9 and LDLR are both similarly associated with LDL cholesterol and coronary artery disease, but LIPC is incorporated into this group – this gene is associated with HDL cholesterol and not with coronary artery disease. As such, whatever criteria linking these genes together is unlikely to be biologically important.

We considered various clustering approaches and noticed some variability depending on 1) the metric used to derive the distance matrix (euclidian, maximum, manhattan,...), 2) the linkage criterion (centroid, mean, ward,...), 3) the input data (absolute vs relative number of hits), and 4) the set of features considered (lipoprotein type, size and class). As a result, for our initial clustering we ended with a naïve clustering based on a visual inspection.

Determining the “right” approach is not trivial, and results might change slightly as new associations are identified. While it seems to us that any methodological choice would be a bit arbitrary, we understand that the reviewer, and some readers, might prefer a more formal approach. Therefore, we took the opportunity of this revision to re-run this analysis using a very standard approach (hierarchical clustering using the centroid method and Euclidian distance) applied to the relative number of hits for the lipoprotein type –i.e. the group with the strongest heterogeneity. The only main change is the [LIPC, PCSK9, LDLR] group of genes being redistributed into other groups –which is in agreement with the reviewer suggestion that LIPC and PCSK9-LDLR potentially belong to

different groups. Figure 4 and Supplementary Figure 6 were updated accordingly and now include the hierarchical cluster. A detailed description of the clustering method is provided in the Supplementary Notes.

Major 2. The authors now incorporate fine-mapping of LIPC for the top metabolites. Presumably, these variants were all previously associated with plasma lipids in larger datasets. Therefore, wouldn't fine mapping of a larger dataset using conventional lipids be more powerful to identify causal variants at LIPC? Do the heterogeneous SNP-metabolite associations for the putative causal SNPs provide any new insight?

Comparing the results from our fine-mapping against one perform for total lipids in a larger dataset would be of interest. However, we believe that a formal analysis would be a full project by itself. Instead our objective for the *LIPC* analysis was to compare results from a simple fine-mapping across all associated metabolite to assess “*the presence of multiple causal variants affecting different metabolites.*”, i.e. addressing the second part of the reviewer comment. To better emphasize this specific point, we moved the detailed description of the cross-reference annotation to the supplement and added a paragraph describing the distribution of posterior probability across the associated phenotypes. The text now reads as:

“Our analysis suggests there are at least 3 distinct association sites with consistently high probabilities of causal effects from 7 SNPs and heterogeneous metabolite association patterns, confirming the likely presence of metabolite-specific variants within this gene (Figure 5 and Supplementary Tables 11-12).

*We cross-referenced top variants of these three sites with GWAS of common human diseases³⁴, and functional annotations from Haploreg³⁵. The first site (A) is composed only of SNP rs10468017, which was previously found associated with age-related macular degeneration (AMD)³⁶⁻³⁸ and with LIPC expression in human liver tissue³⁹. The second site (B) includes 4 SNPs in complete LD that were previously associated with hypertension⁴⁰ and AMD^{41,42}. Among the 4 SNPs, rs2070895 is the strongest candidate in our data with a potential association path through a regulation by *USF1*, a gene with a record of association with lipids⁴³⁻⁴⁶. Finally, the last site (C) included 2 SNPs, among which rs113298164 clearly harboured the highest number of relevant bio-features. It is a rare missense mutation which has been reported to be involved in hepatic lipase deficiency⁴⁷. Additional details on the functional annotation analysis are provided in the Supplementary Notes.*

Deciphering the posterior probability across all SNP-metabolite pair would be challenging because of the dimensionality of the fine-mapping results. However, some global patterns were observed. Overall, large HDL (L_HDL), and triglyceride in lipoprotein (L_HDL_TG, IDL_TG, L_LDL_TG, XL_HDL_TG, LDL_TG, S_LDL_TG, M_LDL_TG, HDL_TG, XS_VLDL_TG) appear to be influenced by all three likely causal sites. Conversely, intermediate-density lipoproteins (IDLs) and several fatty acids (SFA, PUFA, FAW6, TotFA) are likely mostly influenced by sites A and B. Very small VLDL (XS_VLDL) also display heterogeneous posterior probabilities, highlighting mostly variants from site B as likely causal. Finally, while the three sites show the highest posterior probability for most of the metabolites (Supplementary Tables 12), other variants in the region might be involved. For example, an additional variant (rs7177289) displays the strongest posterior probability for the ratio of fatty acids (FAW6_FA, MUFA_FA, LA_FA, and PUFA_FA).”

Major 3. In the prior version, I described the issue with attributing genes to SNPs by proximity, particularly for lipids. A key challenge is the APOE-C1-C2-C4 region. The authors addressed this by clarifying that SNP-gene attributions are by nearest gene. Unfortunately, this still doesn't address this major issue, particularly at this locus. The claims made throughout this study are gene-based. The authors highlight APOC1 at this locus but there are coding SNPs in APOE even more strongly associated with lipids (PMID: 30140000). Also, they attribute the 1p13.3 association to CELSR2 but functional studies attribute the SNP associations to another gene that is not the most proximal gene, SORT1 (PMID: 20686566). Also, they attribute 1p31.3 gene to DOCK7 but there is stronger functional and genetic support for ANGPTL3 (PMIDs: 28538136, 28385496). The current analyses do not permit causal gene claims throughout the abstract, results, and discussion, and the authors may be inadvertently highlighting non-causal genes.

This is obviously a non-trivial question, that in our opinion plagues the entire field – i.e. are the genes reported for association from all existing large-scale GWAS actually the one involved in the outcome analyzed? We would like to have a clear answer to that question, but we can unfortunately only provide a partial answer.

To avoid misleading the readers, we now highlight in the results section that the reported genes are not necessarily the causal one. We support our claim by providing one additional supplementary table presenting the results from a FUMA analysis mapping variants with genes based on their association with gene expression across various tissues. Our updated text says:

“As mentioned previously, those are the nearest genes to the top associated variants for each region. For clarity, we use those genes throughout our study, however, this list should be considered with caution as the genetic effects of the associated variants might potential be attributed to other genes. For example, we performed a bioinformatics analysis using FUMA²⁶, mapping variants with genes based on their association with gene expression. For many of regions, the variants in questions were associated with a range of other candidate genes besides the listed ones (Supplementary Table 8).”

Major 4. The statin interaction analyses allow for unique analyses in METSIM, which is a strength of this study. Since statins strongly influence LDL-C, and particularly non-HDL-C, a helpful secondary analysis would be to adjust for LDL-C change, or non-HDL-C change. In general, key unique aspects of the paper are the statin and age interaction analyses. I would incorporate more of a discussion about these analyses than on power improvements from CMS since the CMS method has already been previously described.

We understand the reviewer interest for assessing the impact of statin. However, we believe that our study is insufficiently powered for the evaluation of complex models involving statin usage. Moreover, the suggestion of adjusting for LDL-C and non-HDL-C changes might raise potential collider bias issues, making the results hard to interpret. Nevertheless, we extended the discussion to highlight better the importance of age and statin in our study as follows:

“The observation of age and statin interactions further highlights the utility of obtaining extensive clinical phenotype data as well as collecting multiple time points which are much better powered to identify age effects that cohort studies. The statin interactions suggest that genetic variation may influence the effectiveness and impact of the drug at a given dose, and may underlie our recent

observation⁵⁵ that statin effects are non-uniform on secondary phenotypes such as fasting glucose across individuals. While the current study is not sufficiently powered to examine these questions directly, it does identify relevant genes to examine for pharmacogenomic studies of statin in properly designed cohorts. The age interactions are also a unique aspect of this work and raise the possibility that genetics can impact trajectories of metabolism over an individual's life span. Although speculative, the most intriguing possibility is that genetic variants could mitigate metabolic disease risk by slowing the natural alteration of metabolic profiles across time."

Minor 1. I'm not sure if there are formatting issues but Supplementary Table 6 seems to have some issues. The column labeled "Kettunen 2016" has various p-values which are copied over to "minimum P" after a dark line but there should be another dark line to the right so it's clear that it doesn't apply to Willer 2013. Also, why is the column for "Willer 2013" empty but there are various p-values corresponding to these SNPs for the different lipid fractions in columns U-X?

We understand our supplementary tables were a little heavy and indeed, the label was incorrect for Table S6. We checked all tables and simplified them whenever possible. Table 3 now reports the 588 significant region-metabolite association along signals previously reported for the same region-metabolite pair. Table 6 reports the novel associations only, along the *p*-value at the exact same SNP and the same metabolite for the 2 studies where data was available. It also includes the results from Willer et al 2013 for the total lipids.

Reviewer #3 #####

On first assessment of the modified manuscript, the authors appear to have worked hard to put their work into proper context with the current literature and address fundamental analytical interpretations of their novel method. Their assessment of major regulators of lipid particle biology perhaps puts into better detail a good deal of known lipid biology, however they provide little insight into the novel associations.

There additional analyses do a reasonable job of attempting to show the variability in discovery is due largely to either power deficits (failure to replicate lower effect signals) and or large power gains (for CMS-only associations). This conclusion is rather bolstered by reasonably good comparison to associations in broader lipid measures studies in much larger sample sizes. I greatly appreciate the deeper insight into the variants discovered by CMS.

We thank the reviewer for his positive feedback on our additional work

I remain concerned that the authors have done an adequate comparison of the literature and publicly available data. For examples, they only list 5 overlapping variants from Davis et al., yet a cursory examination of that publicly available data (<http://csg.sph.umich.edu/boehnke/public/metsim-2017-lipoproteins/>) yields more than 200 overlapping genome-wide significant variants. It was not clear from the tables which of the 247

associations they are claiming to be "novel," but I would be shocked if a good deal of them were not already observed in this earlier (larger) analysis the same data.

Perhaps of equal or more value than the loci these two very similar studies both discovered, is an understanding of the discrepancies between the two. Variability in discoveries from very different study samples is not a major surprise. For both population, cohort, methodological, and chance reasons, loci will differ. However, since these two studies use essentially the same underlying data, it is rather more disturbing not to have more consistency. If there really are 247 novel associations among the 588 presented here, why is the overlap so poor? Similarly, there are dozens (hundreds?) of associations in Davis not observed here. A better understanding of these discrepancies (and a rational explanation for them) would also likely give a better true understanding of the value of the CMS method.

In our comparison with the Davis et al. paper, done at the previous revision stage, we indeed only included in our comparison the few associations that were reported as "new" in the manuscript.

We have now performed a systematic comparison using the complete GWAS summary results from all available metabolites. We applied the same strategy as the one used for Kettunen et al, 2016, i.e. for each and every of the 588 metabolite-region association, we extracted the most significant variant for the same metabolite in a 1MB window around our top SNP for that region. We assume the region is known if it passes the same p-value threshold we used to claim a signal at the discovery stage ($P < 1.28e-9$). Note that there are a few region-metabolites ($N=14$) that had p-value within $[5e-8 ; 1.28e-9]$ that we did not count as it sounded fair to us to use the same threshold for all studies.

Overall, our total count of new associations did not change much: we found that 19 out of the 247 associations were actually also found at the same p -value threshold, so that our total of new association is now 228. It is unclear to us why those 19 associations have not been reported as new ones in Davis et al (e.g. six associations involving ApoB_ApoA1 ratio), as both us and Davis et al did not find them in the literature (i.e. these association were not reported in supp table 4 from Davis et al, which list known associations). We have not tried to investigate this further as this is clearly out of the scope of our study.

Now, to address the second comment about discrepancies: the identification of signals in Davis et al missed by our analysis can easily be explained by 1) a reduced power in general as the sample size in our study is smaller (i.e. the gain achieved by CMS remain limited for many SNPs), and 2) the additional SNP coverage, so that signal in a region might be better captured by SNP not included in our analysis. Therefore, we did not explore these differences further.

Conversely, we spent some time deciphering signals found in our analysis but not in Davis et al (still at the same P-value threshold of $< 1.28e-9$). Among the 588 region-metabolites associations we identified, 351 that were not found by Davis et al. The reasons for discrepancies are multiples:

- 257 corresponds to associations with metabolites not tested by Davis et al (we considered 158 metabolites versus 72 in Davis et al).
- 85 corresponds to associations identified only by CMS, for which the average gain in power corresponded to an effective sample size of approximately 15,000 individuals.
- 9 corresponds to associations identified by both the standard approach and CMS, which might be due to other factors (see below).

While standard linear regression (not adjusting for CMS covariates) showed strong consistency with the Davis et al results, differences for the latter 9 signals might be explained by differences in the analysis setting:

- In Davis, “Traits were adjusted for age, age2, smoking status, and lipid lowering medication. Residuals were inverse normalized”, while we performed a systematic inverse-normal transformation of all phenotypes, and covariates were directly added to the model.
- As discussed by the reviewer in its following comments, we did not adjust for population structure, although this is unlikely to explain much differences as no association was found between top PCs and metabolites (see response to the next comment)
- In our analysis, we restricted for some phenotypes the sample size by removing statin user and fibrate users. This might have resulted in both gain and loss in power, depending on whether the genetic effect differ by statin use.

Numerous papers have shown that even seemingly homogeneous populations such as Finland can exhibit geographical structure. While METSIM is from a small region, I do not think it is appropriate not to do any adjustment for population structure. Some adjustment or justification for not adjusting must be provided. This may explain some of the discrepancies discussed above.

In order to confirm there was no or at least a negligible impact of confounding effect due to population structure, we derived the top 10 principal components of the genotype matrix and tested each of them for association with each of the 158 metabolites considered in our GWAS. Overall, we did not observe any deviation from the null across the 1,580 tests performed. The observed minimum p-value equals 0.0016, which is in line with the expected minimum under the null. The histogram of the p-value (below) did not show either any obvious deviation from the uniform. Therefore, population structure is unlikely to explain any of the discrepancies mentioned by the reviewer.

Reviewers' Comments:

Reviewer #2:

Remarks to the Author:

I appreciate the authors' thorough responses to my concerns. My concerns have been suitably addressed. I have no further comments.

Reviewer #3:

Remarks to the Author:

Thank you for the thorough evaluation of your results in relation to the published literature. This more thorough assessment helps to put the set of findings into proper perspective and the characterization of their analyses makes clear the value of the approaches undertaken in comparison to other approaches.

While I strongly agree with Reviewer 1's concerns about accurately naming causal genes, I am sympathetic to the author's challenge. They are upfront about the challenge, and systematic in their approach.

I have no further concerns.